# TCTP regulates genotoxic stress and tumorigenicity via intercellular vesicular signaling

Robert Amson[1], Andrea Senff-Ribeiro[1,14], Teele Karafin[1,14], Alexandra Lespagnol[1,14], Joane Honoré[1], Virginie Baylot [1], Josette Banroques[2], N Kyle Tanner[2], Nathalie Chamond [3], Jordan D Dimitrov[4], Johan Hoebeke[5], Nathalie M Droin [6], Bastien Job[7], Jonathan Piard[8], Ulrich-Axel Bommer [9], Kwang-Wook Choi [10], Sara Abdelfatah [11], Thomas Efferth[11], Stephanie B Telerman[12], Felipe Correa Geyer[13], Jorge Reis-Filho[13] & Adam Telerman [1✉]

## Abstract

Oncogenic intercellular signaling is regulated by extracellular vesicles (EVs), but the underlying mechanisms remain mostly unclear. Since TCTP (translationally controlled tumor protein) is an EV component, we investigated whether it has a role in genotoxic stress signaling and malignant transformation. By generating a *Tctp*-inducible knockout mouse model (*Tctp*⁻/f ), we report that Tctp is required for genotoxic stress-induced apoptosis signaling via small EVs (sEVs). Human breast cancer cells knocked-down for TCTP show impaired spontaneous EV secretion, thereby reducing sEV-dependent malignant growth. Since *Trp53*⁻/⁻ mice are prone to tumor formation, we derived tumor cells from *Trp53*⁻/⁻;*Tctp*⁻/f double mutant mice and describe a drastic decrease in tumorigenicity with concomitant decrease in sEV secretion and content. Remarkably, *Trp53*⁻/⁻;*Tctp*⁻/f mice show highly prolonged survival. Treatment of *Trp53*⁻/⁻ mice with sertraline, which inhibits TCTP function, increases their survival. Mechanistically, TCTP binds DDX3, recruiting RNAs, including miRNAs, to sEVs. Our findings establish TCTP as an essential protagonist in the regulation of sEV-signaling in the context of apoptosis and tumorigenicity.

Keywords Bystander Effect; Signaling; Small Extracellular Vesicles (sEVs); Tumor Reprogramming; Tumor Reversion
Subject Categories Cancer; Membranes & Trafficking; Signal Transduction

## Introduction

TCTP (also referred to as Tpt1/HRF/Fortilin) is a pro-survival factor (Amson et al, 2013; Bommer, 2017; Telerman and Amson, 2009; Tuynder et al, 2004; Tuynder et al, 2002) with anti-apoptotic activity (Li et al, 2001) that is highly conserved through phylogeny, from unicellular organisms to plants and humans (Brioudes et al, 2010). TCTP contains a BH3-like domain, which upon binding of Bcl-xL, an anti-apoptotic Bcl2 family member, activates its function (Thebault et al, 2016). TCTP also activates Mcl-1, another anti-apoptotic protein (Liu et al, 2005). Moreover, this anti-apoptotic effect of TCTP results from its antagonistic role on p53. We found a reciprocal repression between P53 and TCTP (Amson et al, 2012). Indeed, TCTP promotes the degradation of p53 in a MDM2-dependent manner and p53 represses the expression of TCTP. In line with the latter, *Trp53* null mice express elevated levels of Tctp (Amson et al, 2012), which might explain, at least partially, the spontaneous increase in tumor formation of these mice (Donehower et al, 1992). This hypothesis was not tested before since there was no genetic model available. *Tctp* null mice die in utero at day E7.5. Therefore, the inducible deletion of *Tctp*, would allow us to investigate its effect on tumorigenicity. More specifically, the fact that TCTP is secreted via exosomes and has been suggested to bind proteins involved in RNA transport (Amson et al, 2013; Amzallag et al, 2004; Li et al, 2016), prompted us to investigate its function in intercellular communication, as a potential regulator of sEV secretion and content. Exosomes (Pan and Johnstone, 1983) have been reported to play a pivotal role in cancer by vehiculating oncogenic material (Al-Nedawi et al, 2008) to other cells and in the initiation of the pre-metastatic niche (Costa-Silva et al, 2015; Peinado et al, 2017). We and others have shown that activation of

[1]Institut Gustave Roussy (IGR), Unité Inserm U981, Bâtiment B2M, 114 rue Édouard-Vaillant, 94805 Villejuif, France. [2]Université de Paris Cité & CNRS, Expression Génétique Microbienne, IBPC, 13 rue Pierre et Marie Curie and Institut de Biologie Physico-Chimique, Paris Sciences et Lettres University, CNRS UMR8261, EGM, 75005 Paris, France. [3]Faculté de Pharmacie de Paris, Laboratoire CiTCom - UMR CNRS 8038 Université Paris Descartes 4 Avenue de l'Observatoire, 75270 Paris, France. [4]Centre de Recherche des Cordeliers, INSERM, CNRS, Sorbonne Université, Université de Paris, 75006 Paris, France. [5]Institut de Biologie Moléculaire et Cellulaire, UPR CNRS 9021, 15, rue René Descartes, 67084 Strasbourg, France. [6]Institut Gustave Roussy, Unité Inserm U1287, 114 rue Édouard-Vaillant, 94805 Villejuif, France. [7]Institut Gustave Roussy (IGR), Bioinformatics Core Facility, 114 rue Édouard-Vaillant, 94805 Villejuif, France. [8]Département de Chimie, Ecole Normale Supérieure Paris-Saclay, 4 avenue Des Sciences, 91110 Gif-sur-Yvette, France. [9]Graduate School of Medicine, Faculty of Science, Medicine & Health, University of Wollongong, Wollongong, NSW 2522, Australia. [10]Department of Biological Sciences, Korea Advanced Institute of Science and Technology, Daejeon 34141, Korea. [11]Department of Pharmaceutical Biology, Institute of Pharmaceutical and Biomedical Science, Johannes Gutenberg University, Staudinger Weg 5, 55128 Mainz, Germany. [12]Department of Genetics, University of Cambridge, Cambridge CB2 3EH, UK. [13]Department of Pathology, Memorial Sloan Kettering Cancer Center (MSKCC), 1275 York Ave, New York, NY 10065, USA. [14]These authors contributed equally: Andrea Senff-Ribeiro, Teele Karafin, Alexandra Lespagnol. ✉E-mail: atelerman@gmail.com

p53 by γ-irradiation induces secretion of exosomes through a mechanism involving TSAP6 (Tumor Suppressor Activated Protein), a six-transmembrane domain protein (Amson et al, 1996). TSAP6 is also able to bind and promote the secretion of TCTP via exosomes (Amzallag et al, 2004). The relevance of TCTP as a regulator of exosome secretion and content has not been examined till now. It has been shown that deletion of p53 resulted in a defective exosome secretion upon γ-irradiation, whether experiments were done on normal or cancer cells (Lespagnol et al, 2008; Yu et al, 2006). Extracellular vesicles (EVs) consist of a heterogeneous population consisting of small extracellular vesicles (sEVs) and non-vesicular structures (NVs). Many past and current studies have been performed on these mixed populations because the method to isolate these EVs is straightforward. However, recent work on the composition of EVs redefined the field, providing methodologies that increased accuracy in the isolation of sEVs from NVs using high-resolution iodixanol gradients, together with the availability of new markers (Crescitelli et al, 2021; Gyuris et al, 2019; Jeppesen et al, 2019; Zhang et al, 2018). Given the progress made in the characterization of sEVs, we can now properly address questions about the genetic and molecular way through which TCTP defines the content and function of sEVs. First, we investigated their role in the "Bystander effect" (Azzam et al, 1998; Azzam and Little, 2004), which describes how γ-irradiated cells signal cell death to other cells that have not been irradiated. Then, we characterized another aspect of intercellular communication in a $Tctp^{-/f-}$-specific genetic context, which is the spontaneous secretion of exosomes as described for the transfer of oncogenic material (Al-Nedawi et al, 2008). Altogether, our data demonstrate that TCTP is a key and required regulator of sEV signaling of genotoxic stress (the "Bystander effect") and intercellular transport of oncogenic information.

## Results

### Tctp-regulated sEV-signaling in genotoxic stress-induced apoptosis

In order to explore the role played by Tctp in sEV-signaling, we generated an inducible *Tctp*-knockout mouse model ($Tctp^{-/f}$), the total knockout ($Tctp^{-/-}$) being lethal at day E7.5 (Susini et al, 2008). The strategy to generate the inducible *Tctp*-knockout mice (schematic representation Fig. 1A), is detailed in the Materials and Methods and Appendix Fig. S1. Briefly, a first *Tctp* construct was made containing three LoxP sites and a Neo-cassette (L3-construct). This construct was transfected in embryonic stem cells (ESCs). After selection with neomycin, the surviving clones were tested for homologous recombination, injected into blastocysts, and transferred into mice. Heterozygous mice for this L3-construct, called $Tctp^{+/f}$, were crossed with Cre-mice, and the offspring were treated with tamoxifen to delete the Neo-cassette, preserving the two remaining LoxP sites. The mice containing this second construct (L2) were crossed with mice having the heterozygous constitutive knockout of the *Tctp* gene ($Tctp^{-/+}$) (Susini et al, 2008). The final offspring containing one constitutively deleted *Tctp* allele, the conditional L2 allele, and Cre were named $Tctp^{-/f}$. The genetically modified mice were injected intraperitoneally (IP) four weeks after birth with tamoxifen, which led to a deletion of the

*floxed* (L2) sequence ($Tctp^{-/f-}$) (Fig. 1A). This method provided an efficient ablation of the *Tctp* gene, with as a consequence, a strong decrease in Tctp protein (Fig. 1B). Upon DNA damage, as exemplified by γ-irradiation, p53 function is activated (Lowe et al, 1993). We examined the effect of γ-irradiation of splenocytes from the $Tctp^{-/f-}$ mice and measured an increase in P53 and PSer15-P53 (Fig. 1C), which is in line with our report on *Tctp* heterozygous mice (Amson et al, 2012). Tamoxifen never completely reached all cells in the organism, which explains why Tctp still remained slightly detectable (Fig. 1B,C). While the methodologies to isolate sEVs have improved, quantification and accurate analysis of these specific thymocytes derived vesicles in the serum of mice still requires an extra in vitro step. This is to avoid a mixture of sEVs derived from different organs as well as the necessity to use an excessive number of mice because of the low yield of sEV purification. For this, thymocytes from these γ-irradiated mice were cultured for 24 h. Supernatants were collected and the harvested EVs were further purified by high-resolution iodixanol density gradient centrifugation (Jeppesen et al, 2019), yielding the specific low-density sEV fractions (Fig. 1D). Tctp was detected in the low-density fractions (1 to 6–7), containing the sEVs including exosomes (Amzallag et al, 2004). These low-density fractions were positive for Syntenin-1 and CD81, markers of sEVs, but negative for Fibronectin, which is a marker of the non-vesicular (NV) structures present in the high-density fractions. The isolated sEVs had the expected size, ranging from 40 to 150 nm (Fig. 1E) (Jeppesen et al, 2019). $Tctp^{-/f-}$ thymocytes produced 70% less sEVs than wild-type thymocytes used as control (Fig. 1E,F). In addition, the content of the sEVs was altered upon *Tctp* deletion, as 50% less proteins per $10^7$ sEVs was measured compared to controls, along with the drastic decrease in RNA content (Fig. 1G,H). The uptake by *wild-type (WT)* thymocytes of FITC labeled sEVs derived from either γ-irradiated *wild-type* or $Tctp^{-/f-}$ thymocytes (Fig. 1I,J) was comparable as measured by FACS analysis. Co-culture of reporter cells (*WT* thymocytes) with sEVs derived from the γ-irradiated mice thymocytes led to induction of a strong apoptosis in the reporter cells, as measured by caspase 3/7 activity (Fig. 1K) or TUNEL assay (Fig EV1). In contrast, upon co-culture of reporter cells with sEVs derived from $Tctp^{-/f-}$ thymocytes, significantly lower cell death was quantified (Figs. 1K and EV1). These results indicate that Tctp plays an important role in the secretion of sEVs, in their RNA and protein content, and in vehiculating cell death signaling following genotoxic stress.

### TCTP regulates sEV-dependent malignant growth in human breast tumor cell models

To examine the regulatory role of TCTP in sEV-signaling in the growth of malignant cells, we efficiently generated several short hairpin RNA (shRNA)-mediated knockdown (KD) of *TCTP* in four human breast cancer cell lines, i.e., MCF7, T47D, SKBR3, and MDA-MB231 (Fig. 2A). We investigated the spontaneous secretion of sEVs by these tumor cell lines without γ-irradiation or any other induction that could potentially activate their secretion. sEVs were isolated by high-resolution iodixanol density gradient centrifugation, quantified, and characterized (Fig. 2B–D). The *TCTP* knockdown (using the most efficient shRNA, sh7239) in the different cell lines led to both a reduced number of secreted sEVs and a decrease in their protein and RNA contents (Fig. 2D–F). We then analyzed

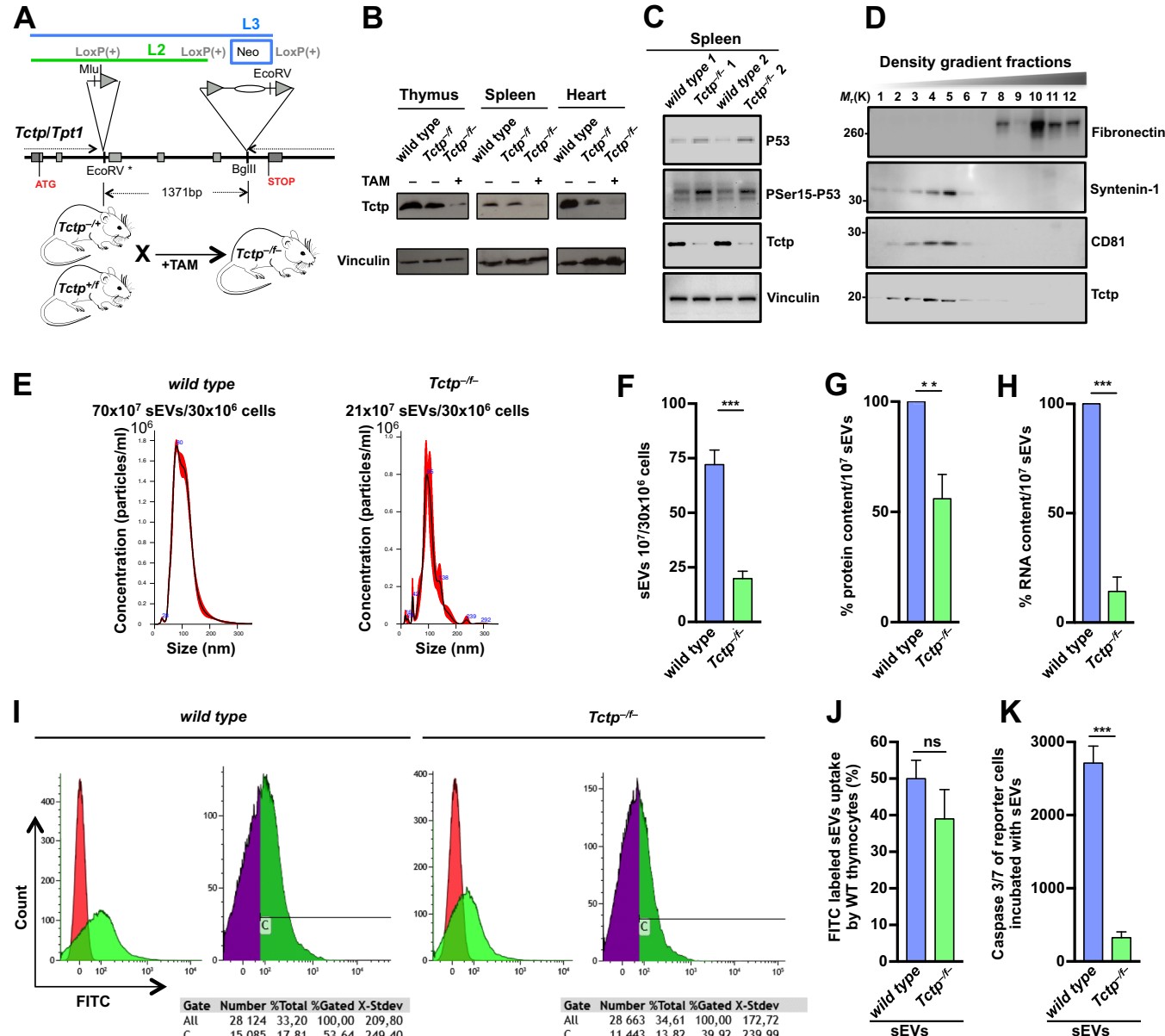

**Figure 1. sEV-signaling following inducible deletion of *Tctp* (*Tctp⁻ᐟᶠ⁻*) in mice.**

(A) Schematic representation including the construct for the mouse mutated *Tctp/Tpt1* allele with the insertion of three LoxP sites, used to generate the inducible *Tctp*-knockout mice (*Tctp⁻ᐟᶠ⁻*). *Tctp⁻ᐟ⁺* are the mice heterozygous for the complete knockout of the *Tctp* gene, the *Tctp⁻ᐟᶠ* are the mice containing the constitutively *Tctp*-knockout and the floxed allele which is deleted after tamoxifen treatment yielding *Tctp⁻ᐟᶠ⁻*. (B) Tctp protein expression in thymus, spleen, and heart from *wild-type* (*WT*) mice and ones bearing one constitutive knockout allele and one inducible allele (*Tctp⁻ᐟᶠ*) before (−) and after (+) tamoxifen (TAM) treatment (*Tctp⁻ᐟᶠ⁻*). Vinculin expression was used to assess equal loading. (C) P53, PSer15-P53, and Tctp protein expression in the spleen of γ-irradiated (10 Gy) *wild-type* and tamoxifen-treated *Tctp⁻ᐟᶠ⁻* mice, Vinculin was used as loading control. (D) Characterization of sEVs after density gradient fractionation of supernatants from cultured thymocytes (γ-irradiated *WT* mice, 10 Gy). After the high-resolution iodixanol gradient, equal volumes of each fraction were loaded on SDS-PAGE gels and membranes were hybridized with the indicated antibodies (Fibronectin, Syntenin-1, CD81, and Tctp). (E) Representative experiments of a Nanoparticle Tracking Analysis (NTA) of vesicles from the above experiment, *WT* (left panel) and tamoxifen-treated *Tctp⁻ᐟᶠ⁻* mice (right panel). These experiments were performed on the pool of the density gradient fractions 1–7 corresponding to the sEV fraction produced by $30 \times 10^6$ thymocytes from γ-irradiated mice. (F) NTA for sEV analysis secreted by thymocytes of γ-irradiated mice ($n = 8$ mice per genotype, blue: *WT*, green: *Tctp⁻ᐟᶠ⁻* mice). (G, H) Quantification of proteins and RNA (relative expression) from $10^7$ sEVs derived from thymocytes of γ-irradiated mice ($n = 8$ mice per genotype). (I) Representative experiment showing the uptake of FITC labeled sEVs by *WT* thymocytes. These sEVs were either derived from γ-irradiated *WT* thymocytes (left panels) or *Tctp⁻ᐟᶠ⁻* thymocytes (right panels). For each genotype: Left graph: thymocytes without any prior incubation with sEVs were used as control (red); thymocytes incubated in the presence of FITC labeled sEVs (green). Right graph: FITC negative population (purple), FITC positive gated population (green) (C gate). Values of the gating are displayed below the graph. (J) Quantification of the same experiment as I, FITC labeled sEV uptake ($n = 8$ γ-irradiated mice per genotype, blue: *WT*, green: *Tctp⁻ᐟᶠ⁻* mice) by reporter *WT* thymocytes. (K) Caspase 3/7 activity in reporter thymocytes co-cultured with sEVs derived from γ-irradiated *WT* or *Tctp⁻ᐟᶠ⁻* mice ($n = 8$ mice per genotype). In each microwell containing 20,000 cells in a volume of 180 µl, $2 \times 10^7$ sEVs were added (as measured by NTA). Data information: All statistical analysis was performed using ANOVA: Mean ± SEM, ns (non-significant), **$P < 0.01$ and ***$P < 0.001$. Source data are available online for this figure.

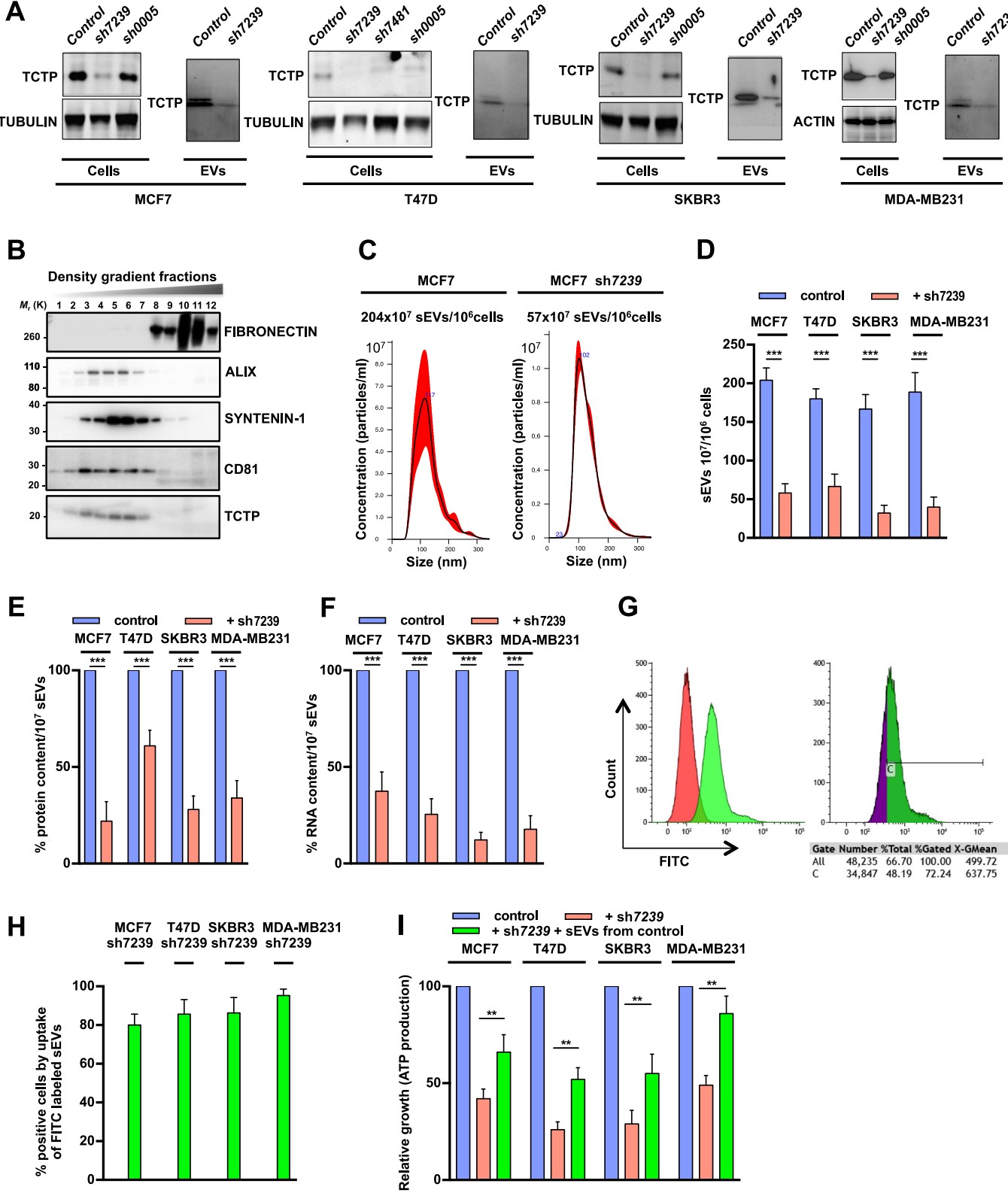

◄ **Figure 2. Knockdown of *TCTP* in human breast cancer cell lines.**

Breast cancer cell lines (MCF7, T47D, SKBR3, MDA-MB231) were knocked-down (KD) for *TCTP* using various shRNAs. (A) Western blot analysis for TCTP expression (left panel: cells) and extracellular vesicles (right panel: EVs) from the respective cell lines (TUBULIN or ACTIN were used as controls for equal loading). (B) High-resolution iodixanol density gradient as exemplified for sEVs derived from MCF7 cells. Western blot analysis of equal volumes of each fraction using the indicated antibodies (against FIBRONECTIN, ALIX, SYNTENIN-1, CD81, and TCTP). (C) Representative NTA experiments for sEVs from MCF7 and MCF7 sh7239 cell lines (production of sEVs by $10^6$ cells). (D) Quantification of sEV production (by $10^6$ cells) using NTA of MCF7, T47D, SKBR3, MDA-MB231 cell lines (parental cells, blue and shTCTP cells, red; $n = 9$ per cell line). (E, F) Relative content of proteins and RNA in $10^7$ sEVs comparing the parental cells (blue) and shTCTP cells (red) ($n = 9$ per cell line). (G) Representative experiments showing the uptake of FITC labeled sEVs derived from parental MCF7 cells by MCF7 sh7239 cells. Left graph: Unlabeled MCF7 sh7239 cells (red), uptake of FITC labeled sEVs (green). Right graph: FITC negative population (purple), FITC positive gated population (green) (C gate). The value of the gating is displayed below the graph. (H) Uptake (% positive) of FITC labeled sEVs originating from the parental cells by sh7239 treated cells. Graph displaying values for sh7239 MCF7, T47D, SKBR3 and MDA-MB231 cells ($n = 3$ per cell line). (I) Relative growth as measured by ATP production of MCF7, T47D, SKBR3, and MDA-MB231 cells (blue). Growth of the same cell lines where TCTP was knocked down with sh7239 (red). Measurement of growth in the complementation experiments in which 20,000 TCTP knocked down cells were supplemented with $200 \times 10^7$ sEVs as measured by NTA isolated from the respective parental malignant cell lines (green) ($n = 8$ per cell line). The readout for the growth was always performed after 4 days of culture. Data information: All statistical analysis were performed using ANOVA, Mean ± SEM **$P < 0.01$ and ***$P < 0.001$. Source data are available online for this figure.

whether the shTCTP cells were still functional in their uptake of sEVs. A representative experiment shows the effective uptake of FITC labeled sEVs derived from the parental breast cancer cells by the respective cell lines treated with sh7239 (Fig. 2G,H; Appendix Fig. S3). Of note, the uptake of FITC labeled sEVs derived from shTCTP cells by shTCTP cells was also highly efficient (sh MCF7 and sh MDA-MB231 cells (Appendix Figs. S4 and S5)), suggesting that the uptake of sEVs is not a TCTP-dependent process.

These *TCTP*-knockdown cells showed a strongly diminished growth. All four sh*TCTP* cancer cell lines regained a robust growth, after being complemented with sEVs from the respective parental cell line (Fig. 2I). These data indicate that TCTP regulates sEV-dependent intercellular signaling of malignant transformation in human breast cancer cell lines. Noticeably, the four cell lines used in this experiment differ in their P53 status. MCF7 cells have a *WT TP53* sequence but are deficient in downstream effectors of apoptosis (Jänicke et al, 2001). T47D cells contain the amino acid substitution C194T in P53, SKBR3 cells bare the R175H substitution in P53, and MDA-MB231 cells have R280K P53 substitution (Muller and Vousden, 2014). The latter three cell lines have a missense mutation in the DNA binding domain of P53 and are gain of function (GOF) mutants. Therefore, this suggests that the effect of the loss of TCTP on sEV-dependent malignant growth, is likely not p53-dependent, when analyzing the spontaneous secretion of sEVs by cancer cells. It is important to note that the experiments here, investigating the spontaneous secretion of sEVs by cancer cells, addressed a different question than those reported previously which investigated the effect of γ-irradiation on exosome secretion (Yu et al, 2006). These findings prompted us to evaluate the consequences of conditional loss of TCTP in *Trp53*$^{-/-}$ driven malignancy and subsequently the role played by sEVs in this process.

### *Trp53*$^{-/-}$ induced tumorigenicity is strongly reduced when *Trp53*$^{-/-}$ mice are crossed with *Tctp*$^{-/f-}$ mice

The prominent role played by p53 in human cancer and the fact that *Trp53*$^{-/-}$ mice are prone to spontaneous tumor formation (Donehower et al, 1992; Lane, 1992), led us to further investigate the genetic link between *Trp53*$^{-/-}$ and *Tctp* in tumorigenicity. *Trp53*$^{-/-}$ knockout mice were crossed with *Tctp*$^{-/f-}$ mice. As mentioned above, *Tctp* was ablated following IP injection of tamoxifen one month after birth, which resulted in a highly significant decrease in Tctp expression in these mice (Fig. 3A). Kaplan–Meier survival analysis of *Trp53*$^{-/-}$ mice (Fig. 3B) showed

that all the mice died of cancer within 29 weeks, which is consistent with the data reported in the literature (Kemp et al, 1994). Crossing of the *Trp53*$^{-/-}$ mice with the *Tctp*$^{-/f-}$ (*Trp53*$^{-/-}$;*Tctp*$^{-/f-}$) shifted the Kaplan–Meier curve to a significantly prolonged lifespan ($p = 0.002$) and, remarkably, some of the mice were still alive after 75 weeks (Fig. 3B). *Tctp*$^{-/f-}$ mice that were not crossed to *Trp53*$^{-/-}$ did not present any tumors nor premature death (Fig. 3B). Quantification of Tctp levels in the liver of *Trp53*$^{-/-}$;*Tctp*$^{-/f-}$ mice showed that not all Tctp was efficiently knocked out (Fig. 3C). This was further substantiated by reinjecting two mice (labeled D and E in Fig. 3B) with tamoxifen IP. There was a 50% reduction in tumor #D after IP injection (Fig. 3D) and tumor #E was almost completely gone (Fig. 3E). While most of the *Trp53*$^{-/-}$ mice died of lymphoma (Fig. 3G), with just a minority of them having sarcoma (Fig. 3F), crossing with the *Tctp*$^{-/f-}$ ones resulted in a majority of the mice having sarcomas (Fig. 3F,H,I). Together these results indicate that *Trp53*$^{-/-}$-induced tumorigenesis is reprogrammable in vivo by inhibiting Tctp expression.

### sEV-regulated malignant transformation signaling is Tctp-dependent

Previous reports indicate that exosomes vehiculate oncogenic intercellular signaling (Al-Nedawi et al, 2008). Since the precise quantification of specific cancer-derived sEVs in mice remains technically elusive, we investigated the role played by Tctp in sEV-mediated malignant transformation signaling in a system derived from the above tumors. To this aim, we developed an ex vivo proxy consisting of a cell line from one of these sarcomas and called it ITR-1 (standing for Inducible Tumor Reversion cell line). These cells were *Trp53*$^{-/-}$, but still contained an intact *floxed Tctp* (*Tctp*$^{-/f}$) allele that was not deleted (Fig. EV2). Upon incubation of the ITR-1 cells with 4-OH-tamoxifen (4-OHT), the *floxed Tctp* cassette was lost, resulting in *Trp53*$^{-/-}$;*Tctp*$^{-/f-}$ cells. The expression of Tctp was strongly reduced after 24–48 h of 4-OHT treatment (Fig. 4A). Supernatants of ITR-1 cells were harvested, and sEV separation was performed by high-resolution iodixanol density gradient centrifugation (Fig. 4B). A significantly diminished secretion of sEVs with a reduced protein and RNA content in the remaining sEVs was found in 4-OHT-treated ITR cells compared to the untreated control cells (Fig. 4C–F). Before conducting complementation experiments to investigate whether sEVs from the non-treated parental cells could induce again a robust growth in the 4-OHT treated cells, we first

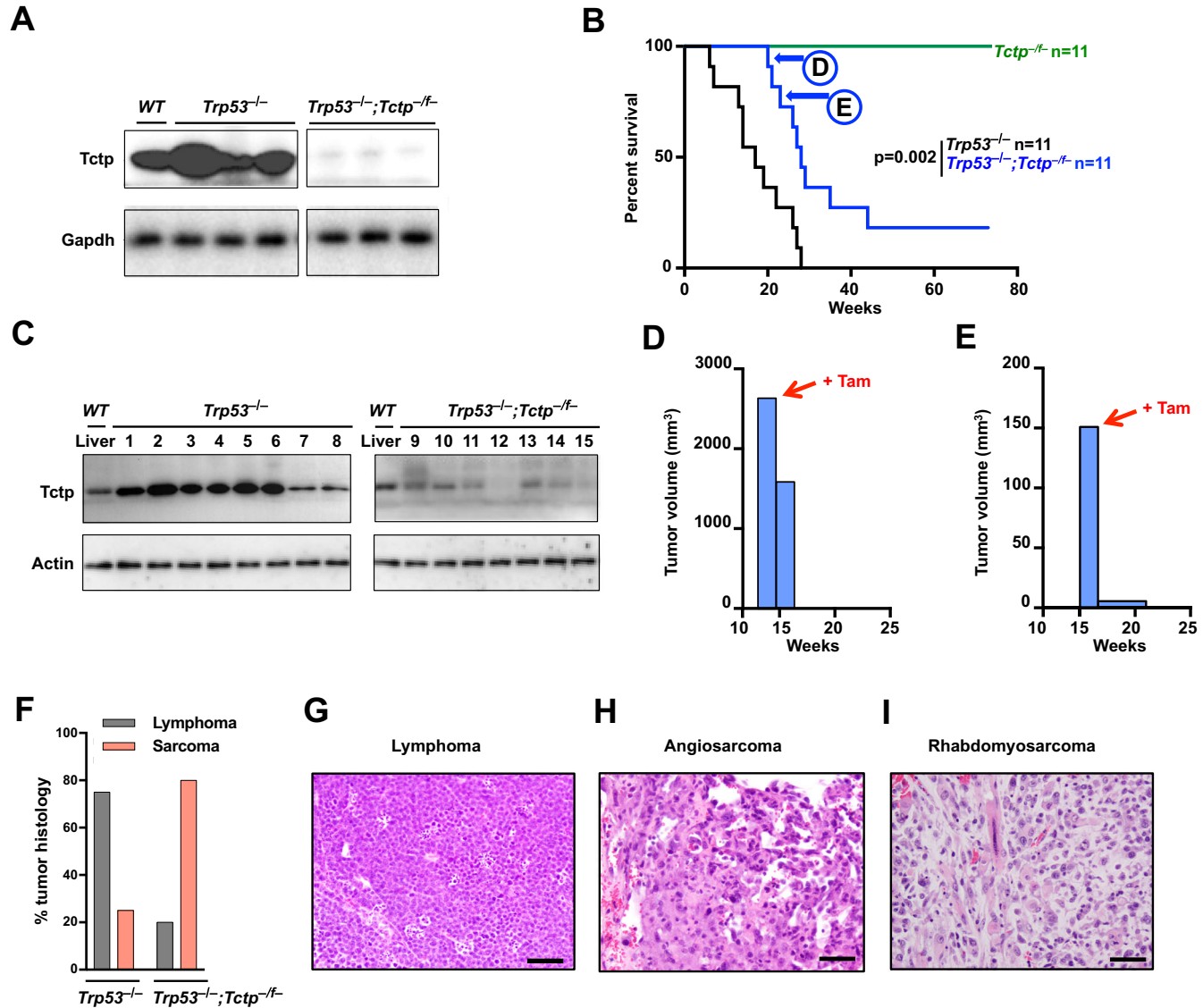

**Figure 3. In vivo reprogramming of *Trp53*^-/-^-dependent tumorigenesis in mice crossed with *Tctp*^-/f-^.**

*Trp53*^-/-^ mice were crossed with *Tctp*^-/f-^ inducible knockout mice to examine the effect of *Tctp* on tumor formation. The crossed animals were treated with tamoxifen four weeks after birth to delete the floxed *Tctp* allele. (**A**) Western blot analysis of Tctp expression in the liver of *WT*, *Trp53*^-/-^ and *Trp53*^-/-^;*Tctp*^-/f-^ mice (Gapdh was used as loading control). (**B**) Kaplan–Meier survival curves of *Trp53*^-/-^ (black), *Tctp*^-/f-^ (green), and *Trp53*^-/-^;*Tctp*^-/f-^ (blue) mice (Wilcoxon test $p = 0.002$), $n = 11$ for each genotype. (**C**) Western blot analysis of Tctp expression in individual tumors displayed in Fig. 3B, the protein extracted from the liver of *WT* mice were used as control (Actin was used as loading control). (**D, E**) Tumor formation in two mice (indicated as D and E in Fig. 3B) that were re-injected with tamoxifen intraperitoneally. (**F**) Histological analysis of tumors derived from *Trp53*^-/-^ and *Trp53*^-/-^;*Tctp*^-/f-^ mice ($n = 8$ for *Trp53*^-/-^ and $n = 10$ for *Trp53*^-/-^;*Tctp*^-/f-^ mice). A majority of lymphomas are found in *Trp53*^-/-^, whereas the proportion is inverted for the *Trp53*^-/-^;*Tctp*^-/f-^ derived tumors towards sarcomas. (**G–I**) Examples of the histopathology of the different tumor subtypes found in the *Trp53*^-/-^;*Tctp*^-/f-^ mice. Scale bars: 100 μm. Data information: (**B**) Wilcoxon test $p = 0.002$, $n = 11$ for each genotype. Source data are available online for this figure.

measured the uptake of sEVs by 4-OHT treated cells. Indeed, the uptake of FITC-labeled sEVs originating from the parental ITR-1 cells, by the 4-OHT-treated ITR-1 cells, was highly efficient (Fig. 4G,H).

The relative growth of the 4-OHT-treated ITR-1 cells compared to control ITR-1 was significantly reduced (Fig. 4I). Rescue experiments indicated that ITR-1 cells treated with 4-OHT and then co-cultured with the parental sEVs (not depleted in *Tctp*), regained a robust growth (Fig. 4I). We can observe an increase in

the expression of TCTP in ITR-1 cells that were treated with 4-OHT and supplemented with sEVs from untreated parental ITR-1 cells (Appendix Fig. S6). These findings suggest that the sEVs do uptake TCTP, which is consistent with the results obtained from the experiments involving FITC-labeled sEVs (Fig. 4G). Neither EVs nor sEVs derived from 4-OHT treated ITR-1 cells, thus depleted of TCTP, could restore the malignant growth (Fig. EV3A–C) despite the fact that the uptake of these sEVs remained highly efficient (Fig. EV3B). These rescue experiments

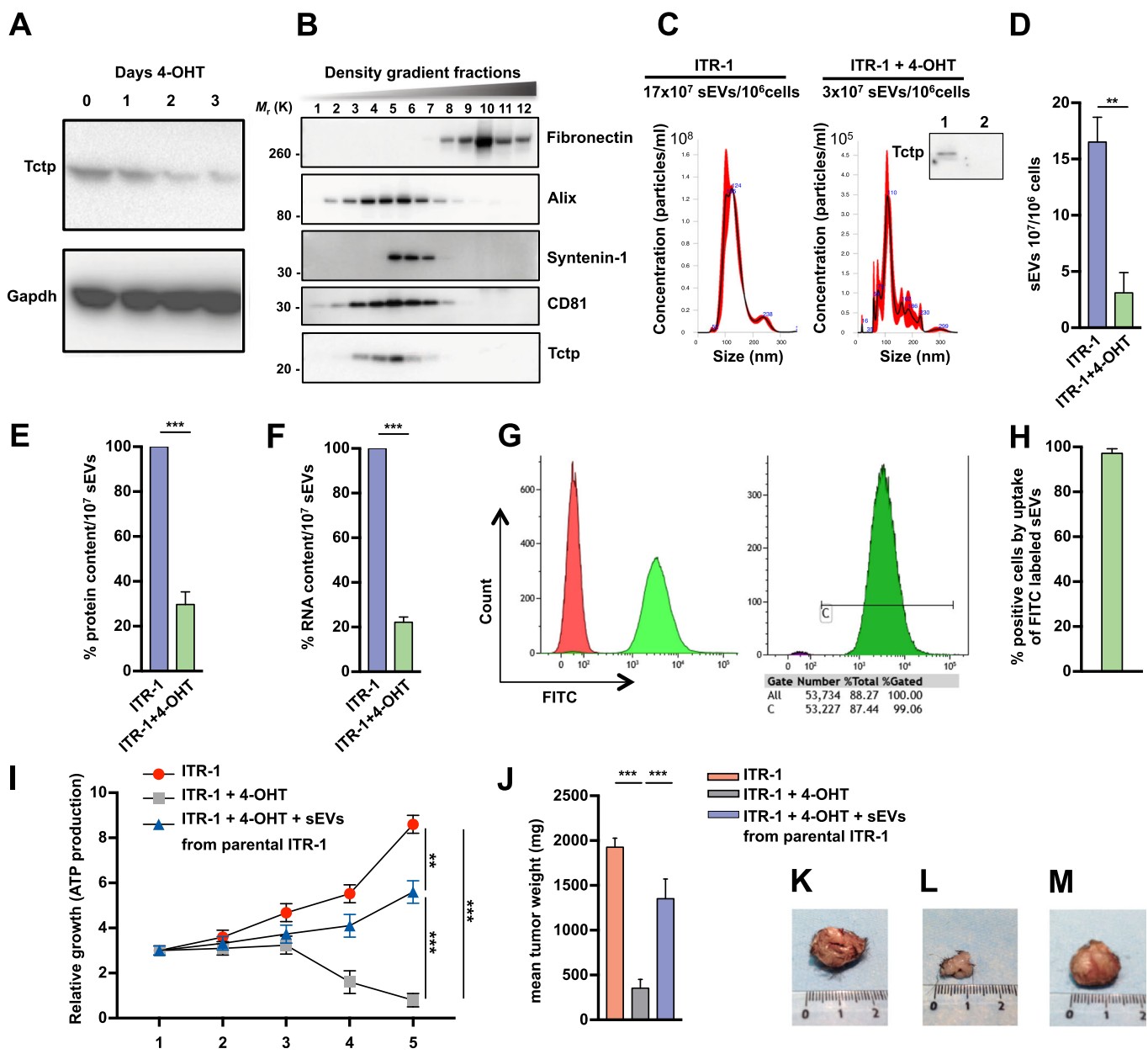

were also highly reproducible in vivo (Fig. 4J–M). *WT* mice injected with ITR-1 cells formed rapidly large tumors (Fig. 4J,K). *WT* mice injected with ITR-1 cells depleted of *Tctp* (following 4-OHT treatment) formed significantly smaller tumors (Fig. 4J,L). Ultimately, *WT* mice injected with ITR-1 cells treated with 4-OHT, but supplemented with the sEVs from the parental ITR-1 cells, formed large tumors again (Fig. 4J,M). These experiments using an ex vivo proxy (ITR-1 cells) indicate that sEVs transfer of oncogenic material is regulated in a Tctp-dependent manner.

## The mechanism by which TCTP regulates the content of miRNAs in sEVs is DDX3-dependent

In a large-scale study aimed at identifying potential RNA binding proteins (Castello et al, 2012), using cross-linking immunoprecipitations (CLIP), cellular TCTP was found to be a potential candidate. Of note, TCTP contains structural features similar to that of the double-stranded RNA binding domain of the DExD/H-box helicase LGP2 (Amson et al, 2013). To explore the mechanism through which Tctp recruits RNAs, we started with CLIP experiments in EVs derived from thymocytes originating from γ-irradiated *WT* mice. Tctp co-immunoprecipitates RNAs in EVs (Fig. 5A). The specificity of this reaction was shown by incubation with two different concentrations of T1 RNAse: only at low concentration of RNAse (0.04 U/μl), a smear corresponding to the precipitation of RNA could be visualized with Tctp but not with control IgG. Following CLIP, RNA was eluted and analyzed by miRNA microarray-hybridization.

Upon a total of 1075 hybridization signals (Dataset EV1), 484 were considered as significant (the cut-off was determined at a

Figure 4.  sEV-dependent tumor formation in an inducible *Trp53^-/-;Tctp^-/f-* cellular system.

A cell line (ITR-1: Induced Tumor Reversion-1 cell line) was derived from a sarcoma of a *Trp53^-/-;Tctp^-/f-* mouse in which not all cells responded to the tamoxifen treatment. Indeed, these sarcoma cells are *Trp53^-/-* with an inducible *Tctp* deletion upon 4-OH Tamoxifen (4-OHT) treatment resulting in *Trp53^-/-;Tctp^-/f-* cells. Hence it is important to note that initially, when derived from the tumor, these cells did not efficiently delete the *floxed Tctp* gene after the initial IP tamoxifen treatment of the mouse. (A) Western blot analysis of Tctp expression in ITR-1 cells treated with 4-OHT for different time intervals (Gapdh was used as loading control). (B) Iodixanol density gradient isolation of sEVs derived from ITR-1 cells. Western blot analysis of equal volumes of each gradient fraction using the indicated antibodies against Fibronectin, Alix, Syntenin-1, CD81, and Tctp. (C) Representative experiment of NTA of ITR-1 isolated sEVs before (left panel) or after 4-OHT treatment (right panel). Inlay: Tctp expression in 170 × 10^7 sEVs of ITR-1 cells before (lane 1) and after 4-OHT treatment (lane 2). (D) Quantification of sEV secretion by ITR-1 cells before (blue) and after treatment with 4-OHT (green) (independent biological replicates n = 9). (E, F) Relative protein or RNA content respectively, in 10^7 sEVs from ITR-1 cells before (blue) and after treatment with 4-OHT (green) (independent biological replicates n = 9). (G) Representative experiment showing the uptake of FITC labeled sEVs derived from parental ITR-1 cells by 4-OHT treated ITR-1 cells. Left graph: Unlabeled 4-OHT treated ITR-1 cells (red), uptake of FITC labeled sEVs derived from the parental ITR-1 cells by 4-OHT treated ITR-1 cells (green). Right graph: FITC negative population (purple), FITC positive gated population (green) (C gate). The value of the gating is displayed below the graph. (H) Uptake (% positive) of FITC labeled sEVs originating from the parental ITR-1 cells by 4-OHT treated ITR-1 cells (independent biological replicates n = 5). (I) Complementation experiments (in vitro): Growth of ITR-1 cells before (red), after treatment with 4-OHT (gray) and the latter following addition of sEVs from the parental ITR-1 cells (blue). For 20,000 cells cultured, 17 × 10^7 sEVs were added (independent biological replicates n = 8). (J) In vivo tumor formation for respectively ITR-1 cells (red) injected subcutaneously in mice (10^6 cells per site), ITR-1 cells treated with 4-OHT (gray) and ITR-1 treated with 4-OHT and supplemented with sEVs (blue) from parental ITR-1 cells (n = 7 mice for each group) (10^6 cells in a volume of 100 μl and supplemented with 850 × 10^7 sEVs were injected per site). Mice were sacrificed after 12 days. (K–M) Examples of tumors from mice injected with ITR-1 cells (K), ITR-1 cells treated with 4-OHT (L), and 4-OHT treated ITR-1 cells supplemented with sEVs from parental ITR-1 cells (M). Data information: (D, E, F, H, J) statistical analysis was performed using ANOVA, Mean ± SEM, or (I) Mann–Whitney test, **$P < 0.01$ and ***$P < 0.001$. Source data are available online for this figure.

signal intensity of 5.950). Among these signals, 65.7% were enriched after anti-Tctp by the CLIP. One should always be careful with the interpretation of these results, particularly for the miRNAs that were not enriched by the CLIP (some being false negatives). We show some examples (Fig. 5B) of these false negatives (let-7c-5p and let-7i-5p) as well as miRNAs that were the most enriched, with all of them being confirmed by other studies to be present in sEVs (Jeppesen et al, 2019). These false negatives are just exemplifying that in the CLIP, Tctp is not binding to the entire amount of the miRNAs present in the extract, because this kind of immunoprecipitation is not highly efficient. These results were further documented in sEVs using qRT-PCR on RNA derived from ITR-1 cells treated or not with 4-OHT. The 4-OHT leads to a deletion of *Tctp* in the ITR-1 cells. mmu-miR-23a-5p was used as reference miRNA and mmu-miR-22-3p as a negative control (Fig. 5C). These data show that deletion of *Tctp* in the sEVs led to a significant decrease in a majority of the miRNAs, where some become almost undetectable. Thus, let-7c-5p and let-7i-5p were Tctp-dependent (Fig. 5C and Dataset EV4). These data indicate that Tctp formed directly or indirectly complexes with different miRNAs in sEVs. Given that a single miRNA can target the expression of dozens to hundreds of genes, it was not possible to find a unique biological pathway corresponding to enriched population binding to Tctp protein (Datasets EV2 and EV3). The purpose of these experiments was not to provide a landscape corresponding to miRNAs binding directly or indirectly to Tctp, but rather to use it as a proxy in order to assess the potential recruitment of RNAs by Tctp. The only reasonable conclusion we can draw from the CLIP analysis is that Tctp is implicated in the enrichment of a wide range of miRNAs which are involved in multiple biological processes ranging from development, apoptosis to cancer (Datasets EV2 and EV3).

To investigate the direct binding of RNAs to TCTP, two methods and different RNA species were used. In an isothermal titration calorimetry (ITC) assay with TCTP and total RNA, no binding could be detected (Fig. 5D). We previously showed that we could readily measure RNA binding with DEAD-box proteins (Banroques et al, 2011). Here, we found that while Ded1, the yeast homolog of DDX3, bound the RNA01 oligonucleotide, no

detectable binding of TCTP with this RNA species was detected (Fig. 5E). We concluded that TCTP, at least under these experimental conditions, was unable to directly bind RNA.

We then hypothesized that TCTP might bind RNA via an RNA-binding protein such as DDX3, which was indicated as a potential TCTP partner (Li et al, 2016) and is also present in sEVs (Jeppesen et al, 2019). We assessed the direct binding of DDX3 to TCTP by Surface Plasmon Resonance (SPR)-based technology (Biacore). A significant affinity ($K_D = 68.9$ nM) (Fig. 5F) was determined for the interaction between these two proteins (Fig. EV4). To investigate whether at least some of the microRNAs pulled down by the anti-TCTP CLIP are also binding DDX3, we analyzed a public database of the microRNAs binding DDX3 (Huang et al, 2022). Out of the 128 microRNAs binding to DDX3 (Huang et al, 2022), 43 of them were found in our anti-TCTP CLIP (Dataset EV1). This suggests that TCTP is recruiting these microRNAs via DDX3. These 43 microRNAs are highlighted in yellow (Dataset EV5).

Previous studies showed that TCTP interacts with TSAP6 by using different technologies (yeast two-hybrid, co-IP, co-localization by IHC) (Amzallag et al, 2004; Lespagnol et al, 2008; Yu et al, 2006) hence we used it as a positive control for the SPR experiments. We confirmed that TCTP binds TSAP6 with an affinity of ~1 μM, validating this methodology and indicating that the affinity of TCTP to DDX3 is higher than TCTP to TSAP6 (Appendix Fig. S2).

Further analyses showed that TCTP did not bind directly to let-7c-5p, but both, DDX3 *wild-type* and DDX-DQAD (a Walker B motif mutant that forms more stable complexes with RNA because the ATP is not hydrolyzed) formed complexes with let-7c-5p (Fig. 5G,H). AMP-PNP (Adenosine 5'-(β,γ-imido)triphosphate) is a non-hydrolyzable analog of ATP, where the β,γ-phosphoanhydride oxygen is replaced by a nitrogen. However, the valence of nitrogen is not the same as for oxygen, which results in the normally linear β,γ-phosphoanhydride bond being kinked. We previously found that yeast Ded1 binds AMP-PNP with the same affinity as ATP, and that it enhances the binding affinity of RNA to around 40–50 nM. However, mammalian DDX3 apparently does not bind AMP-PNP as well, which results in a weaker EMSA signal.

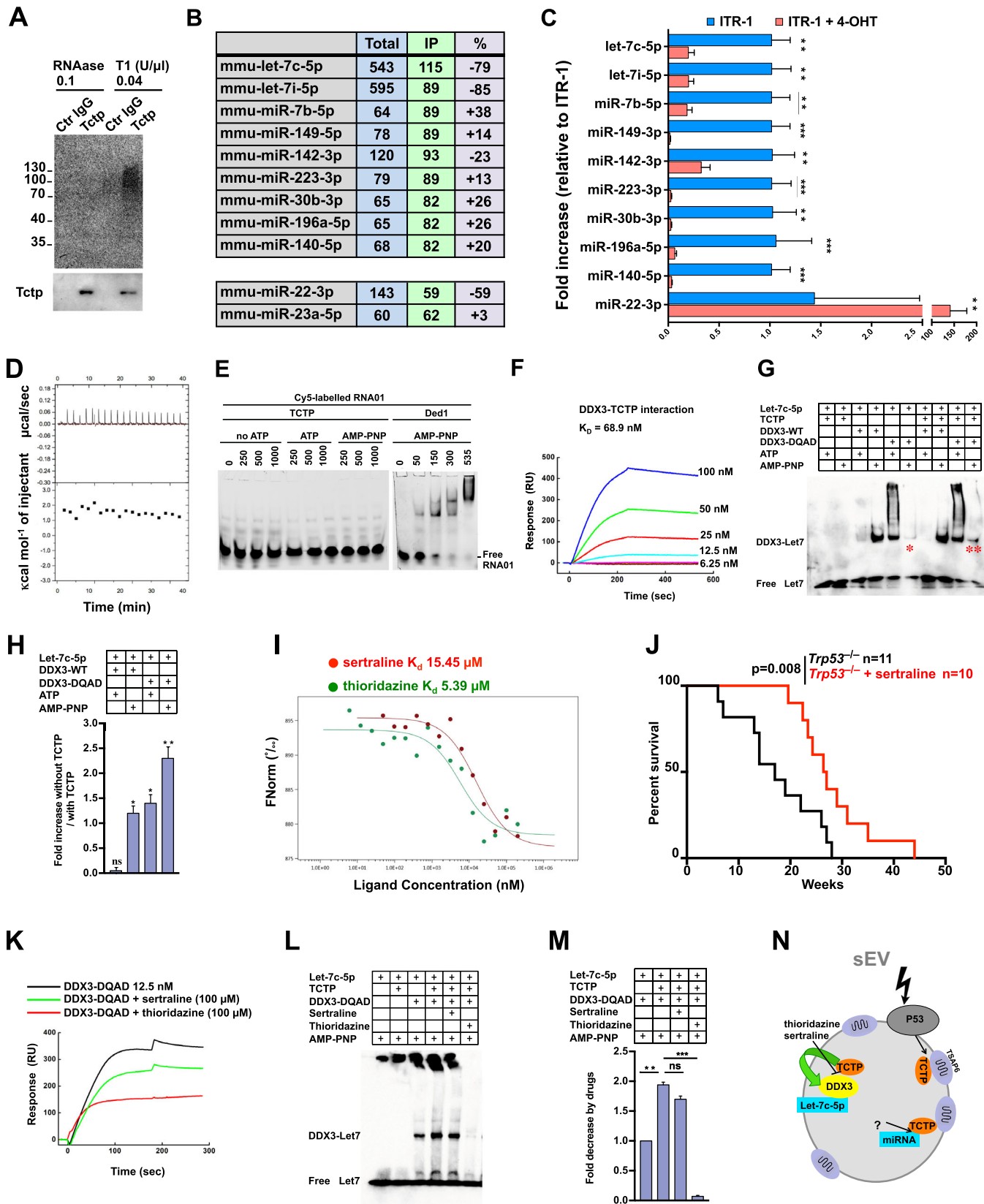

**Figure 5. TCTP forms complexes with DDX3 in recruiting miRNAs.**

(A) Cross-linking immunoprecipitation (CLIP) of EVs from γ-irradiated (10 Gy) thymocytes (WT mice) with either control IgG or anti-Tctp antibodies using two different concentrations of T1 RNase. The smear in the last lane indicates the binding of RNA to Tctp. Lower panel: TCTP co-immunoprecipitation, Western blot analysis with anti-TCTP antibodies. (B) Table displaying ten examples of miRNAs intensity in the microarray-hybridization (fluorescence intensity expressed in arbitrary units), before (Total EV RNA), after the CLIP (IP), with the percentage of enrichment (last column). (C) qRT-PCR analysis: differentially expressed miRNAs in sEVs isolated by iodixanol gradient centrifugation, from ITR-1 cells before (blue) and after 4-OHT treatment (red). mmu-miR-22-3p was used as a negative control. They were normalized to mmu-miR-23a-5p and expressed as the $2^{-\hat{}\hat{}Ct}$ normalized to ITR-1 control (independent technical replicates $n = 3$). (D) Isothermal Titration Calorimetry (ITC) of RNA - TCTP binding. (E) To analyze in a different system a potential TCTP-RNA binding, increasing quantities of TCTP, in nM, was incubated with 50 nM of Cy5-labeled RNA01 (5′-CY5-UCAUACUUUUCUUUUCUUUUCCAUC-3′) at pH 7.5 and separated under non-denaturing conditions on a 6% polyacrylamide gel run at 4 °C. The gel was subsequently scanned with a Typhoon FLA9500 phosphoimager (GE Healthcare) to reveal the labeled RNA. The yeast DEAD-box helicase Ded1 was used as a positive control. AMP-PNP (adenylyl-imidodiphosphate) is a non-hydrolyzable analog of ATP. (F) Surface Plasmon resonance analysis: Real-time interaction profiles obtained after injection of serial dilution of human DDX3 to TCTP immobilized on sensor surface. DDX3-TCTP interaction $K_D = 68.9$ nM. (G) The recombinant proteins were incubated on ice, separated on a non-denaturing gel, transferred to a Hybond™-N+ membrane. Binding of biotinylated let-7c-5p miRNA (50 nM) to TCTP (500 nM), DDX3-WT (300 nM), DDX3-DQAD (300 nM) in the presence of ATP or AMP-PNP. Enhancement of the binding of DDX3-DQAD to let-7c-5p in the absence (one red star) or presence (two red stars) of TCTP. (H) Quantification of the binding of let-7c-5p in the absence of TCTP compared to its binding in the presence of TCTP (independent technical replicates $n = 6$). (I) Binding of sertraline and thioridazine to TCTP analyzed by microscale thermophoresis. Different concentrations of sertraline and thioridazine ranging from 0.01 to 200 µM (1:1 dilution) were titrated against constant concentration of TCTP (785 nM). (J) Kaplan–Meier survival curves of $Trp53^{-/-}$ mice that were treated with sertraline *per os* (PO), 30 mg/kg (body weight) 5 days a week, starting at the age of 6 weeks, compared to $Trp53^{-/-}$ that did not receive the treatment. Sertraline treated $Trp53^{-/-}$ mice ($n = 10$), untreated $Trp53^{-/-}$ mice ($n = 11$). (K) Surface Plasmon resonance analysis of the binding of DDX3-DQAD (12.5 nM) to TCTP in the presence of the vehicle (black), 100 µM sertraline (green) or 100 µM thioridazine (red). (L) Binding of biotinylated let-7c-5p (50 nM) to DDX3-DQAD (300 nM) in the presence of TCTP (500 nM), AMP-PNP and 100 µM sertraline or 100 µM thioridazine. (M) Relative quantification of the binding of let-7c-5p in the absence or presence of TCTP supplemented with sertraline or thioridazine (independent technical replicates $n = 3$). (N) Schematic representation of the interactions between TSAP6, TCTP, DDX3 and let-7c-5p or other single-stranded RNAs. The green arrow shows the potentiation of the binding of DDX3 to let-7c-5p by TCTP. Sertraline and thioridazine inhibit the binding of TCTP to DDX3. Data information: Statistical analysis was performed using (C) Student's t-test, Mean ± SEM, independent technical replicates $n = 3$; (H, M) ANOVA, Mean ± SEM, ns (non-significant), *$P < 0.05$, **$P < 0.01$ and ***$P < 0.001$; (J) Wilcoxon test $p = 0.008$. Source data are available online for this figure.

We then verified whether TCTP might influence the binding of DDX3 to let7c. Using AMP-PMP, a non-hydrolyzable analog of ATP, our data indicate that TCTP potentiates the binding of DDX-DQAD with let-7c-5p (red star * Fig. 5G,H). The DQAD mutant changes the glutamic acid into a glutamine, where the interactions involving the β,γ-phosphoanhydride bond involve a nitrogen in place of an oxygen in the Walker B motif (motif II). Thus, the protein binds ATP but has trouble hydrolyzing the bond or releasing the products. The stereochemistry of glutamic acid and glutamine otherwise seems similar, at least DDX3 and Ded1 are not perturbed (Xiol et al, 2014). Quantification of the let-7c-5p signal showed that by adding TCTP an increase of approximately 50% for DDX-DQAD- let-7c-5p was detected in the presence of ATP while 200% in the presence of AMP-PNP (Fig. 5H). These results indicate that TCTP can recruit miRNA in an indirect way by forming complexes with DDX3.

To further validate this mechanism at the pharmacological level, we used two agents, sertraline and thioridazine, that bind and inhibit the function TCTP (Amson et al, 2012). Microscale thermophoresis (MST) was performed to assess the interaction between sertraline and thioridazine and recombinant human TCTP (Fig. 5I). The method was performed as previously described (Fischer et al, 2021a; Fischer et al, 2021b). Different concentrations of sertraline and thioridazine ranging from 0.01 to 200 µM (1:1 dilution) were titrated against a constant concentration of TCTP. A concentration-dependent fluorescence of TCTP against sertraline and thioridazine was observed (Fig. 5I) with a significant binding of thioridazine to TCTP with a $K_d$ value of 5.39 ± 2.1 µM (standard error of regression). Sertraline was also bound to TCTP with a $K_d$ value of 15.45 ± 1.4 µM (standard error of regression). These results are in line with our previous findings (Amson et al, 2012), confirming the binding of both drugs to TCTP, using different technologies. Since sertraline inhibits the function of TCTP, we tested whether

$Trp53^{-/-}$ induced tumorigenicity was affected by sertraline treatment. Indeed, the survival of these mice was significantly prolonged ($p = 0.008$) (Fig. 5J), mimicking the above genetic results of Tctp deletion.

Given the above results, we investigated the effects of sertraline and thioridazine on the TCTP-DDX3 complexes. Thioridazine, and to a lesser degree sertraline, disrupted TCTP-DDX3 complex formation (Fig. 5K). We then tested the complex formation between TCTP-DDX and let-7c-5p (Fig. 5L) in the presence or absence of each of the pharmacological agents. Sertraline did not display any significant effect. On the other hand, thioridazine disrupted almost entirely the complex formation (Fig. 5L,M). The control experiments measuring the binding of TCTP to TSAP6 indicate that both sertraline and thioridazine are efficient in inhibiting this complex (Appendix Fig. S2).

In summary, these data indicate that TCTP does not directly bind RNA under these experimental conditions, but rather binds to DDX3 and potentiates the latter to bind miRNAs such as let-7c-5p (Schematic representation Fig. 5N). Drugs inhibiting the function of TCTP, disrupt TCTP-DDX3 complex formation and most importantly, sertraline mimics the deletion of Tctp in $Trp53^{-/-}$ tumorigenesis in mice.

## Discussion

In the present study, we aimed to understand at the genetic and molecular level the role of TCTP in intercellular signaling by means of sEVs. More specifically, we investigated this in the context of genotoxic stress and in cancer.

Ionizing irradiation resulting in thymocyte cell death requires a functional P53 (Lowe et al, 1994; Lowe et al, 1993). It has been well documented that genotoxic stress following γ-irradiation and DNA damage induces cell death, not only in the irradiated cells but also

in neighboring or distant cells (Azzam and Little, 2004; Azzam et al, 1998). This is best known as "the Bystander effect" (Azzam and Little, 2004) which became textbook knowledge. Many of these studies were performed by low irradiation, and it has been observed that the results of this irradiation also impacted cells that had not been irradiated (Azzam and Little, 2004; Azzam et al, 1998). Hence, we use the term "Bystander effect" in its broader sense when cells go into apoptosis without themselves being directly irradiated. Several secreted factors have been implicated in this process (Azzam and Little, 2004). The molecular mechanism of the Bystander effect and the role p53 plays herein remain to be elucidated. Previous data indicate that some of the p53 target genes encode for secreted proteins (Buckbinder et al, 1995), which could influence other cells or the microenvironment. It was also found that p53 partially regulates the secretion of extracellular vesicles through TSAP6, which binds TCTP and is also responsible for its secretion by exosomes (Amzallag et al, 2004; Lespagnol et al, 2008; Yu et al, 2006). Here, we show that sEVs secreted by γ-irradiated thymocytes from *WT* mice transferred the apoptotic signal to reporter cells. In contrast, γ-irradiated thymocytes from *Tctp*-inducible knockout mice (*Tctp$^{-/f-}$*), in which both *Tctp* alleles are deleted, are deficient in the signaling of cell death to reporter cells (Fig. 1). Loss of *Tctp* has as a consequence a significantly decreased secretion of sEVs (approximately 1/3 compared to *WT*). Moreover, the sEVs' content was considerably altered, with a decrease in proteins and almost undetectable RNA levels. Thus, we demonstrated that p53-regulated sEV-signaling, following γ-irradiation, is Tctp-dependent. This provides the molecular framework for the Bystander effect and the non-cell-autonomous role of p53 in communicating cell death to other cells, solving some of the problems elaborated on the p53 pathway and exosomes (Levine et al, 2006). This is also in line with the antagonistic effect between p53 and TCTP (Amson et al, 2012). Indeed, to avoid degradation by MDM2, a process that is stimulated by TCTP, p53 represses TCTP in the cells, as reported here by secreting it out of the cell by sEVs. The term "vesicle externalization" was used in the characterization of exosomes (Pan and Johnstone, 1983). The above results, together with the findings that exosomes transfer oncogenic material and initiate the metastatic niche (Al-Nedawi et al, 2008; Costa-Silva et al, 2015; Peinado et al, 2017), prompted us to examine the role of TCTP in sEV signaling of tumor cells. More specifically, the experiments in the present study assessed the spontaneous secretion of sEVs by cancer cells without any induction by γ-irradiation or any other stimulus. This is the most suitable way to examine the transfer by these vesicles of oncogenic information (Al-Nedawi et al, 2008).

We knocked down *TCTP* in four breast cancer cell lines (MCF7, T47D, SKBR3, MDA-MB231). The consequence was a significant reduction in the secretion of sEVs, concomitant with a decrease in protein and RNA contents per vesicle. The loss of TCTP expression led to a significant reduction in the growth of these cells. Importantly, their growth was rescued by adding the sEVs from the parental cells to the *TCTP*-knockdown cell lines. sEVs from the *TCTP*-sh cells were deficient in restoring the tumor growth even when added in high concentration (R. Amson, unpublished observations). These results show that TCTP in the sEVs plays a major role in the intercellular transmission of tumorigenic material, providing a genetic framework for the previously published results (Al-Nedawi et al, 2008).

Having generated inducible *Tctp$^{-/f-}$* mice, we assessed whether deletion of *Tctp* could change the outcome of *Trp53$^{-/-}$* driven tumorigenesis (Fig. 3). We found that *Trp53$^{-/-}$;Tctp$^{-/f-}$* mice had a strongly prolonged survival. This was further confirmed in *Trp53$^{-/-}$* mice by a pharmacological approach using sertraline, an inhibitor of TCTP.

This inducible ablation of *Tctp* and its consequences on tumor formation in *Trp53$^{-/-}$* mice is probably due to a series of events regulated by Tctp (Bommer and Telerman, 2020). In the context of the present study, to address the question of whether Tctp in the sEVs could at least be partially responsible for this reprogramming of tumors, we had to use an ex vivo proxy. Hence, the complementation experiments using the inducible *Tctp*-knockout cell line ITR-1, showed that these *Trp53$^{-/-}$* cells that lost their malignancy as a result of deleting both *Tctp* alleles, regained it upon supplementation with sEVs from the malignant parental ITR-1 cell line (Fig. 4).

Since the question was raised that following deletion of *Tctp* there could be a potential pro-oncogenic signaling that would shift from the initially observed sEVs to other EVs, we investigated whether EVs (containing both the NVs and sEVs) derived from 4-OHT treated ITR-1 cells (depleted in TCTP) could rescue the malignant growth; we did not observe any rescue.

These results provide evidence that *Trp53$^{-/-}$* driven tumorigenesis, ex vivo, is reprogrammable/reversible by ablation of the Tctp-sEV circuitry. It is important to note that in homeostasis, the secretion of sEVs by normal thymocytes is almost undetectable without stimulation. However, as exemplified here upon γ-irradiation, the secretion of sEVs in these thymocytes is p53-dependent. The situation is different in cancer cells where sEVs are secreted on a continuous basis without requiring any stimulus, including γ-irradiation. It should be noted that some cancer cells with wild type p53 function can be induced by genotoxic stress to secrete exosomes (Yu et al, 2006). On the contrary, the spontaneous secretion of sEVs by cancer cells, as examined in the present work, is not likely p53-dependent, whether wild type p53, loss or gain of function mutants. This led us to the conclusion that TCTP in sEV signaling is required and determines the fate of recipient cells regardless of whether it is apoptosis or malignant transformation.

In every experiment described in this study, whether using the inducible *Tctp*-knockout mouse model, the *TCTP* knockdown cells, or the inducible *Tctp*-knockout ITR-1 cells, there was always a drastic reduction in the RNA content of the sEVs.

Hence, we questioned whether Tctp does bind RNAs either directly or indirectly (Fig. 5). The CLIP approach indicated that anti-Tctp antibodies pulled down more than 60% of the miRNAs examined. Some of the most expressed miRNAs were highly downregulated in the ITR-1 cells induced for *Tctp* loss, as shown by qRT-PCR. Tctp participated in the transport of miRNAs, not directly but by complexing to the RNA-binding protein DDX3 (with a $K_D = 68.9$ nM).

Altogether, the data presented indicate that deletion of Tctp has as a consequence a decreased number of sEVs secreted per cell, concomitant with a decrease in protein and RNA content. At least part of the transport of miRNAs by TCTP is mediated as a complex with DDX3. Many of these miRNAs could affect cell growth. However, it is important to take into consideration the fact that in the sEVs, many of these miRNAs are present together with other RNA species and proteins, some of which are RNA binding

proteins, suggesting that the outcome of the phenotype could be a coordinated response of these different RNAs and proteins (Lewis et al, 2005; O'Day and Lal, 2010). Some of them are directly related to DDX3 (Ni et al, 2020; Zhao et al, 2016). Of note, out of the 128 microRNAs binding to DDX3 (Huang et al, 2022), (Dataset EV5), 43 of them were found in our anti-TCTP CLIP.

DDX3 forms complexes with miRNA let-7c-5p and TCTP potentiates this interaction. We previously found that cofactors enhance the RNA-dependent ATPase activity of Ded1 presumably by acting like chaperones (they are not chaperones though, by current definitions) to facilitate the folding of the protein into an active conformation. The flanking N- and C-terminal sequences of Ded1 (and DDX3) have extensive regions of intrinsic disorder, and it tends to aggregate and precipitate, particularly when concentrated. The interactions of the cofactors may help Ded1 assume a functional conformation, although they don't directly stimulate the activity (Senissar et al, 2014). Both sertraline and thioridazine inhibit the binding of TCTP to DDX3, with thioridazine being more efficient. We have previously found that sertraline and thioridazine inhibit the function of TCTP (Amson et al, 2013). Interestingly, our data show that these drugs inhibit the binding of TCTP to TSAP6 (Appendix Fig. S2). Thus, both drugs bind to TCTP and inhibit its function. Together with the in vivo effect of sertraline on spontaneous tumor formation in p53 knockout mice, these results might provide a new rationale in the treatment of cancer. Thioridazine targeting of TCTP also disrupts the complex TCTP-DDX3-let-7c-5p. DDX3 is an RNA helicase which has been implicated in cancer biology, also defining the prognosis of the disease (Chao et al, 2006; Mo et al, 2021; Phung et al, 2019).

With more evidence of the ever-growing complexity of the non-coding RNA networks in cancer (Anastasiadou et al, 2018), it is important to improve our understanding of their transport, and more specifically by sEVs. There is compelling evidence for the transport of different RNA species in exosomes (including mRNA and miRNAs) (Skog et al, 2008; Valadi et al, 2007). It was recently demonstrated that the heterogeneous nuclear ribonucleoprotein A2B1 (hnRNPA2B1) binds specifically to miRNAs and regulates their loading into exosomes (Villarroya-Beltri et al, 2013). One issue is that in some of these studies, the EVs have not been fully characterized, and it is not clear whether they can, with certainty, be considered "sEVs". LC3-conjugating machinery, an autophagy component, was also identified as a regulator of exosome secretion (Leidal et al, 2020). Recent studies have also shown the relevance of small RNA loading in extracellular vesicles in plants (He et al, 2021).

Intercellular transmission of oncogenic material (Al-Nedawi et al, 2008) and pre-metastatic niche initiation (Costa-Silva et al, 2015; Hoshino et al, 2015) are just some of the examples that have made exosome biology an expanding field where genetic models are needed to deepen our knowledge. Our study provides evidence, using genetic, biochemical, and biophysical approaches, that TCTP is required for sEV-signaling of programmed cell death in genotoxic stress in thymocytes and in the transmission of oncogenic material regulating tumorigenesis. Furthermore, given the risk of exosomes from cancer cells transferring oncogenic material to healthy cells (Al-Nedawi et al, 2008; Keklikoglou et al, 2019; Wrighton, 2019), drugs such as thioridazine and sertraline could be a starting point for generating new treatment strategies.

# Methods

## Cell lines

MCF7, T47D, SKBR3 and MDA-MB231 cell lines were purchased from ATCC.

## Mice strains

The heterozygous $Tctp^{-/+}$ mice (C57BL/6) were previously described (Susini et al, 2008), they were generated by deleting $Tctp$ in the embryonic stem cells (ESCs). Only the heterozygous mice were viable. The complete $Tctp$-knockout mice died at E7.5. Therefore, we had to generate an inducible $Tctp$-knockout model (see below strategy) (crossed with $Cre$ mice) in which the gene floxed $Tctp$ gene was deleted one month after birth by injecting tamoxifen intraperitoneally (IP) (see below). $Trp53^{-/-}$ mice were generated by Donehower and co-workers (Donehower et al, 1992).

## Animal experiments

All experimental protocols and animal experiments described in this paper were approved by the ethics committee of CNRS, INSERM, and the Institut Gustave Roussy, Villejuif, France. All animal studies were carried out in the restricted facilities provided by the CNRS, INSERM, and Institut Gustave Roussy Villejuif, France, by authorized personnel following the prescribed rules.

## Strategy used for the generation of inducible $Tctp$-knockout mice ($Tctp^{-/f-}$)

The inducible $Tctp$-knockout mice were generated as illustrated in the main text (Fig. 1A), based on the same construct (detailed previously (Susini et al, 2008)) that was used for the $Tctp$-total knockout. After homologous recombination in ESCs, the $Tctp$ gene was "replaced" by the inducible allele L3 (as shown below for the constructs, the genotyping strategy, and the PCR products in the gel). Correctly targeted ESCs clones were obtained (Appendix Fig. S1A–C), as verified by Southern blot (Appendix Fig. S1D) and PCR. In a second step the Neo-cassette was deleted resulting in the allele L2 that still contains two LoxP sites (Appendix Fig. S1E–H). The Cre-Ert2 gene (Tamoxifen dependent) was inserted in the Rosa 26 gene and under control of the Rosa 26 promotor. PCR controls are using "Rosa-Cre" primers as documented in Appendix Fig. S1I.

The mice homozygous for the floxed allele did not yield completely satisfactory results in wiping out entirely $Tctp$ after tamoxifen treatment, for unknown reasons. Therefore, we always crossed mice with one constitutively knockout allele (Susini et al, 2008) and the second one floxed, yielding mice where $Tctp$ could efficiently be silenced by injecting tamoxifen IP one month after birth ($Tctp^{-/f-}$) (Appendix Fig. S1I).

## Tumorigenesis and survival of $Trp53^{-/-}$ and $Trp53^{-/-}$; $Tctp^{-/f-}$ mice

$Trp53^{-/-}$, $Tctp^{-/f}$ and $Cre$ mice were crossed in order to obtain $Trp53^{-/-};Tctp^{-/f-}$ mice. In order to delete the floxed sequence ($Tctp^{-/f-}$), 4-week-old mice were treated IP with tamoxifen (75 mg tamoxifen/kg body weight) once every 24 h for a total of

5 consecutive days. Mice were observed daily and tumors developed spontaneously, as described previously for $Trp53^{-/-}$ mice (Kemp et al, 1994). Mice were sacrificed following the ethical guidelines and autopsied. Histological analysis was performed on the tumors.

## Tumorigenesis and survival of $Trp53^{-/-}$ mice treated with sertraline

$Trp53^{-/-}$ mice were treated with sertraline *per os* (PO), 30 mg/kg (body weight) 5 days a week, starting at the age of 6 weeks. Sertraline was dissolved in a saline solution pH 7.4 (600 µg/200 µl per day).

## Induced tumor reversion-1 cell line (ITR-1)

This primary cell line was generated from a sarcoma of a $Trp53^{-/-};Tctp^{-/f-}$ mouse by disrupting the tumor, followed by cell culture and selection for the clones using the same PCR strategy as the one used for the inducible *Tctp*-knockout mice. It should be noted here that these ITR clones originating from a sarcoma have both the constitutively deleted *Tctp* and still the floxed *Tctp* gene. Induction of *Cre* was achieved with 5 µM of 4-OH-tamoxifen (4-OHT) with as consequence the deletion of the floxed *Tctp* gene (Fig. EV2).

## Stable shRNA for *TCTP* in human breast cancer cell lines

MCF7, T47D, SKBR3, MDA-MB231 were knocked-down for *TCTP* using various shRNAs:

sh*7239*: GCACATCCTTGCTAATTTCAA
sh*0005*: CTATTGGACTACCGTGAGGAT
sh*7481*: GCAGGAAACAAGTTTCACAAA

Control: The MISSION pLKO.1-puro non-target shRNA control transduction particles contain an shRNA insert that does not target any known genes from any species.

## Protein expression analysis

Proteins were separated on 4-12 Bis-Tris Gel (Invitrogen NP0335BOX, NP0336BOX) using MOPS SDS Running Buffer (NuPAGE™ NP0001). Transfer of the proteins on PVDF membranes (Novex^R PB9320) was performed at 4 °C, 30 V in Tris Glycine Ethanol (20%) Buffer (BioRad 1610734) overnight. Membranes were blocked with Blocking Buffer (SuperBlock™ 37537) for 10 min at room temperature.

Western blot analysis was performed by hybridizing with rabbit anti-TCTP antibodies (1/1000) generated against the full-length TCTP. The polyclonal antisera were affinity-purified on columns coupled with the protein (AgroBio) (Susini et al, 2008). Antibodies used for protein detection in sEVs were rabbit anti-TCTP (human protein) or anti-Tctp (mouse protein) (1/750) (Abcam ab133568), rabbit anti-Syntenin-1 (1/1000) (Abcam ab133267), rabbit anti-Alix (1/1000) (Abcam ab186429), mouse anti-CD81 (1/200) (Santa Cruz sc7637) and rabbit anti-Fibronectin (1/1000) (Abcam ab2413).

Quantification of the loading was normalized using mouse anti-Vinculin antibodies (1/300) (Sigma Chemical V4505), mouse anti-Actin (1/1000) (Santa Cruz sc-8432), mouse anti-α Tubulin (B7) (1/1000) (Santa Cruz sc-5286), or mouse anti-GAPDH (1/1000)

(Santa Cruz sc-47724). As secondary antibodies, mouse anti-rabbit IgG HRP (1/5000) (Jackson Immuno 711-035-152) or donkey anti-mouse IgG HRP (1/5000) (Jackson Immuno 984320) were used.

The hybridization was performed using standard procedures. Blots were processed with Super Signal™ using a Molecular Imager Camera (BioRad).

## sEV isolation and analysis

Cell culture supernatants were centrifuged at $300 \times g$ for 5 min, $2000 \times g$ for 20 min, then at $15,000 \times g$ for 40 min and $120,000 \times g$ for 3 h (Jeppesen et al, 2019; Thery et al, 2006). The obtained EVs were purified by high-resolution (12–36%) iodixanol density gradient fractionation (Jeppesen et al, 2019) to yield the sEVs. The recovered fractions were centrifuged at $120,000 \times g$ for 3 h, resuspended in 100 µl of RPMI 1640 without fetal bovine serum (FBS) for co-culture experiments. For RNA isolation, sEVs were resuspended in 350 µl Trizol (Invitrogen). For protein analysis, they were resuspended in 100 µl RIPA buffer.

## Quantification and particle size distribution of extracellular vesicles by Nanoparticle Tracking Analysis (NTA)

Each sEV sample was diluted (1/5 to 1/200) to comply with the precision of the Nanosight measurements. Samples were loaded using a syringe-pump into a Nanosight NS300 (Malvern) configured with a 405 nm laser and a high sensitivity sCMOS camera at 23 °C. Data were analyzed with a NTA 3.3 software (Nanosight) with a detection threshold of 4. The graphs represent the averaged finite track-length adjusted (FTLA) plot of the concentration (particle/ml) as a function of particle size (nm) averaged from three captures (1 min each).

## Evaluation of sEV yield following high-resolution iodixanol gradient centrifugation

In order to quantify the final concentration of sEVs by high-resolution iodixanol gradient, we first measured by NTA the number of EVs, corresponding to the $120,000 \times g$ pellet. We then compared the number of sEVs and NVs obtained after high-resolution iodixanol gradient to the initial $120,000 \times g$ EV count. This is exemplified for the production of EVs, sEVs and NVs derived from γ-irradiated *WT* mice thymocytes (Fig. EV5).

## FITC labeling of sEVs and cellular uptake

FITC (Sigma-Aldrich, cat. no. F3651) was dissolved in ethanol at 20 mg/ml. sEVs were incubated with 1 mg/ml FITC in 1 ml PBS containing 0.05 M $NaHCO_3$ and $Na_2CO_3$ (pH 8.5) for 7 min in the dark at RT (Lundberg et al, 2016). Unbound FITC was removed by washing the sEVs in 30 ml PBS and ultracentrifugation at $120,000 \times g$ for 3 h. Those sEVs were resuspended in 500 µl RPMI, supplemented with 10% EV-depleted FBS and added to reporter cells (consisting of *wild-type* thymocytes originating from 2-month-old C57BL/6 mice) in a final volume of 5 ml. After overnight incubation, the cells were washed three times and analyzed by FACS. The same procedure was performed for the cancer cell lines. Acquisition was performed with a LSR Fortessa cytometer (BD

Biosciences) equipped with BD Diva software. Data were analyzed using the Kaluza software (Beckman Coulter).

## Co-culture of sEVs and reporter cells, measurements of apoptosis, and tumor cell growth

sEVs from thymocytes were isolated from γ-irradiated mice and added to the reporter cells, consisting of thymocytes from non-irradiated *wild-type* mice. Cells were seeded in 96-well plates ($20 \times 10^3$ cells per well in complete medium depleted of vesicles). The sEVs were incubated with the reporter cells overnight. The same way sEVs isolated from the parental cell lines (MCF7, T47D, SKBR3, MDA-MB231) were added to the *shTCTP* (sh7239) knockdown cells up to 5 days. sEVs isolated from ITR-1 cells were added to the ITR-1 (*Tctp*-inducible knockout cells) treated with 4-OHT. As additional control sEVs purified from 4-OHT treated ITR-1 cells were added to 4-OHT treated ITR-1 cells. The apoptosis for the reporter thymocytes was measured by either Caspase-Glo® 3/7 Assay (Promega) or Click-iT™ Plus TUNEL Assay following the manufacturer's instructions (Invitrogen). The growth of the knockdown (MCF7, T47D, SKBR3, MDA-MB231) or knockout ITR-1 4-OHT cells, following the addition of sEVs, was measured using quantification of ATP production using Celltiter-glo (Promega).

## Cross-linking immunoprecipitation (CLIP) followed by a miRNA microarray-hybridization

Thymocytes from γ-irradiated *wild-type* mice were cultured overnight and EVs were isolated using standard procedures (Lespagnol et al, 2008). CLIP was performed using formaldehyde. Briefly, after isolation and purification, EVs pellets were gently transferred to an Eppendorf tube and incubated in 0.03% methanol-free formaldehyde in PBS for 10 min at room temperature to crosslink. Then, the reaction was quenched by the addition of 125 μl glycine (2 M) and incubated for 5 min at room temperature, washed once by centrifuging at $22,000 \times g$ for 15 min at 4 °C with PBS containing protease and RNase inhibitors. The pellet was lysed in 400 μl RIPA supplemented with protease, and RNase inhibitors by vortexing and keeping on ice for 20 min. The lysate was sonicated $3 \times 5$ s on ice, pausing for 1 min between each burst using the Sonics Vibra-Cell Cell Disrupter at 20% amplitude. To the lysate, 50 μl of protein G beads coupled to either 10 μg of anti-TCTP (Abcam rabbit polyclonal ab37506) or anti-Vinculin antibody (Santa-Cruz sc-5573) were added in a total volume of 400 μl RIPA supplemented with protease. The IP mixes were incubated on a wheel at 4 °C overnight. The beads were washed once with ice-cold 400 μl RIPA supplemented with protease, twice with RIPA and 1 M urea, incubating 10 min at room temperature before centrifugations, again twice with RIPA and 1 M urea and once with ice-cold RIPA. The crosslinks were reverted by the addition of 130 μl of Reverse Buffer (10 mM Tris-HCl pH 6.8, 5 mM EDTA, 10 mM DTT, 1% SDS) supplemented with RNasin by incubating at 70 °C for 45 min with gentle shaking. An aliquot of 700 μl Qiasol was added to the reaction and total RNA was isolated with the miRNeasy mini kit (Qiagen).

## miRNA expression analysis (microarray) and ranking comparison

EVs RNA was analyzed before and after CLIP. Each sample was prepared according to the Agilent miRNA Microarray system

protocol (Agilent Technologies, Inc.). RNA from EVs was labeled with pCp-Cy3 using T4 RNA ligase (GE Healthcare Europe GmbH) and hybridized to Agilent G3 Mouse miRNA Microarray Release 16.0, $8 \times 60$ K (G4872A-031184; Agilent Technologies, Inc.), according to the manufacturer's protocol. All processing methods used for miRNA analyses were performed on the Cy3 Median Signal in FeatureExtraction software v10.7 (Agilent Technologies, Inc.). Raw data files were extracted using base R functions in BioConductor. Control probes, spots flagged by FeatureExtraction and spots corresponding to probes invalidated by the manufacturer were systematically removed, then intensity data were log2-transformed. Probe-level normalized values were aggregated to miRNA-level using their median value. miRNAs with more than 30% of missing values across all samples were discarded. Missing values for resulting miRNAs were imputed using k-nearest-neighbor.

After finding that the non-normalized expression for the IP sample was much more condensed than samples of other categories, we resolved to not perform the commonly used quantile-normalization across samples. To assess the IP/Total class comparisons, we computed the median rank of each miRNA for each sample class, and ordered this ranked list by decreasing median rank of the IP class. We affected the median log2-transformed and direct-intensity values of each miRNA for each class, and computed the percentage enrichment of IP over the Total class.

## miRNA expression quantification

Murine miRNA expression was analyzed after reverse transcription in a final volume of 45 μl using a Taqman microRNA Reverse Transcription Kit and Real-time quantitative PCR with Taqman detection protocol in a 7500 thermocycler (Applied Biosystems, Thermo Fisher Scientific). Taqman RT primers and primers/probes were: let-7c-5p #000379, let-7i-5p #002221, miR-7b-5p #000378, miR-142-3p #000464, miR-149-5p #002255, miR-223-3p #002295, miR-30b-3p #000602, miR-196a-5p #241070, mmu-miR-140-5p #001187, mmu-miR-877-3p #002548 and miR-22-3p #000398. They were reported to miR-23a-5p #002439 (all from Applied Biosystems, Thermo Fisher Scientific). They were expressed as the $2^{-\wedge\wedge Ct}$ normalized to ITR-1 control.

## RNA-protein binding assays

RNA binding assays were conducted as previously described (Banroques et al, 2008). Cy5-labeled RNA01 (5′ CY5-UCAUAC UUUUCUUUUCUUUUCCAUC 3′) and the Cy5-labeled miRNAs let-7c-5p (5′ Cy5-UGAGGUAGUAGGUUGUAUGGUU 3′) were obtained from Eurofins Genomics. Free and bound RNAs were separated under non-denaturing conditions on a 6% polyacryla-mide gel run at 4 °C. The gels were subsequently scanned with a Typhoon FLA9500 phosphoimager (GE Healthcare) to reveal and quantify the labeled RNA. The yeast DEAD-box helicase Ded1 was used as a positive control; it has a high, ATP-dependent, RNA binding activity. Reactions were incubated without nucleotide, with 5 mM ATP or with 5 mM adenylyl-imidodiphosphate; (AMP-PNP; Sigma). We tested the binding of biotinylated let7c-5p (50 nM) to TCTP (500 nM), DDX3-WT (300 nM), or DDX3-DQAD (300 nM). TCTP was pre-incubated with either DDX3-WT or DDX3-DQAD

on ice for 20 min. The binding assay was performed in the presence of ATP or AMP-PNP for 25 min on ice. Free and bound RNAs were separated under non-denaturing conditions on a NativePAGE™ 3–12 Bis-Tris Gel % (Invitrogen BN1001BOX) using NativePAGE™ Running Buffer (Invitrogen BN2001) and pre-run at 4 °C, 100 × V during 30 min and run at 4 °C, 100 V. The transfer on Amersham Hybond™-N+ membranes was performed overnight at 4 °C and 30 V in NuPAGE$^R$ Transfer Buffer (NP0006-1). Detection of biotinylated let7c-5p using the LightShift™ RNA EMSA Optimization and Control Kit was performed following the instruction of the manufacturer (Thermo Scientific 20158X). To assess the effect of 100 μM sertraline or 100 μM thioridazine on the binding of biotinylated let-7c-5p (50 nM) to DDX3-QAD (300 nM) in the presence of TCTP (500 nM), TCTP was pre-incubated with either drug for 15 min on ice.

## Isothermal titration calorimetry experiments

Titrations were done by isothermal titration calorimetry (ITC) assays using an ITC200 calorimeter from GE-Microcal. TCTP was used at 0.065 mM in 50 mM bicarbonate ammonium pH 9. RNA isolated from irradiated *wild-type* thymocytes was used in the same buffer. Titrations were carried out at 24 °C. Data were analyzed using Microcal Origin software provided by the ITC manufacturer (GE-Microcal).

## Complementation experiments of ITR-1 cells with sEVs in vivo

The growth of ITR-1 cells in vivo was assessed by subcutaneous injection of $10^6$ cells on both the right and left sides of *wild-type* mice. Alternatively, a second group of *wild-type* mice were injected with $10^6$ ITR-1 cells that underwent treatment with 5 μM 4-OHT. A third group of *WT* mice were injected with $10^6$ ITR-1 cells treated with 5 μM 4-OHT and complemented with $850 \times 10^7$ sEVs. All mice were sacrificed after 12 days, autopsied and tumors were weighed.

## Surface plasmon resonance analysis for protein interaction

Real-time kinetic analyses were performed by using surface Plasmon resonance-based system Biacore 2000 (Biacore, Uppsala, Sweden). TCTP was immobilized on CM5 sensor chips (Cytiva, Biacore, Uppsala, Sweden) by standard amino-coupling procedure. In brief, a mixture containing 50 mM N-hydroxisuccinimide and 200 mM N-ethyl-N'-dimethylaminopropyl carbodiimide (GE Healthcare Biacore) was injected over the carboxymethylated dextran surface of the chip for 4 min with a flow rate of 10 μl/ min. For covalent binding to the activated surface TCTP was diluted to a final concentration of 10 μg/ml in solution of 5 mM maleic acid (pH 3.5). The protein was injected with flow rate of 10 μl/min and achieved immobilization levels for TCTP was 697 RU. Blocking of the unreacted carboxyl groups was achieved by injecting 1 M ethanolamine-HCl for 4 min. The immobilization procedure and further binding analyses were performed with HBS-EP (10 mM HEPES pH 7.2; 150 mM NaCl; 3 mM EDTA, and 0.005% Tween 20) as running buffer which was filtered through 0.22 μM filter and degassed under vacuum.

To evaluate the binding kinetics and affinity of the interaction with TCTP, DDX3-WT, DDX3-DQAD were serially diluted (two-fold each step) in HBS-EP to concentrations ranging from 0.195 to 100 nM. The proteins were then injected over the immobilized TCTP. The flow rate during injection of both was 30 μl/min. The association and dissociation phases of the interactions were monitored for 4 and 5 min, respectively. For regeneration of the sensor chip 3 M solution KSCN was injected for 30 s contact times. Kinetics measurements were performed at a temperature of 25 °C.

The evaluation of the kinetic data was performed by BIAevaluation version 4.1.1 Software (Biacore) by applying global Langmuir binding with drifting baseline model to the experimental data.

## Microscale thermophoresis analysis of sertraline and thioridazine

Microscale thermophoresis (MST) was performed for assessment of the interaction between sertraline, thioridazine, and recombinant human TCTP. The method was performed as previously described (Fischer et al, 2021a; Fischer et al, 2021b). The TCTP protein was labeled using Monolith Protein Labeling Kit RED- NHS 2nd Generation (MO-L011, NanoTemper Technologies GmbH, Munich, Germany) according to the manufacturer's instructions. TCTP protein at 785 nM was titrated against different concentrations of sertraline and thioridazine. The following analysis buffer was used: 50 mM Tris buffer pH 7.0, 150 mM NaCl, 10 mM MgCl$_2$ and 0.05% Tween 20. Samples of the interacting components were filled into Monolith NT.115 standard capillaries (MO-K022, NanoTemper Technologies GmbH, Munich, Germany). Monolith NT.115 instrument (NanoTemper Technologies) was used for fluorescent signal measurement. The test was performed using 20% LED power and 10 MST power. For analysis, we used MO. The fitting curve of interaction and calculation of dissociation constant ($K_d$) was performed with Affinity analysis software (Nano Temper Technologies).

## Statistical analysis

Statistical analysis of results presented in Figs. 1, 2, 4, 5: error bars were expressed as ±SEM (standard error of the mean). Statistical analysis was performed using ANOVA or t-test *$P \leq 0.05$, **$P \leq 0.01$, and ***$P \leq 0.001$ were considered significant. Statistical analysis of the Kaplan–Meier survival curves (Figs. 3 and 5) was performed using the Wilcoxon and Log-rank test. Mann–Whitney tests were performed on the growth curves of ITR-1 cells (Figs. 4 and EV3). No blinding was performed.

# Data availability

No primary datasets have been generated and deposited.

# Peer review information

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

## Acknowledgements

AT and RA dedicate this work to Susan M. MacDonald in recognition for her superb work, sharing always the experimental data, for her constant support and help. We thank Christelle Martin, Karelia Lipson, Catherine Cailleau, Aude Queixalos, René Papion-Duchateau at the animal facility, SEAT/CNRS, for their help in generating the *Tctp*-inducible knockout mice. Alexandre Diet, Marie-Laure Dessain and Emilie Bouvier at TAAM-Phenomin and Ayma Galland at the animal facility of Université Paris-Saclay for taking care of the mice colonies. We thank Christopher M. Johnson (LMB, University of Cambridge) for advice and help with the ITC analysis and Cyril Catelain and Philippe Rameau for their expertise in FACS analysis. This work was supported by grants from INCa Projets libres 2013-1-PL BIO-10-IGR-1 « Biologie et Sciences du Cancer » 2013, the French National Agency for Research (ANR, ANR-09-BLAN-0292) and LabEx LERMIT to A.T., and Odyssea fund to RA. ASR was supported by a fellowship from Science without Borders (CNPq), CAPES-COFECUB and Federal University of Parana (UFPR, Brazil). TK was supported by a doctoral fellowship from MESR, Ecole Doctorale de Cancérologie Paris XI. The synopsis image has been created with BioRender.com.

## Author contributions

**Robert Amson**: Conceptualization; Data curation; Formal analysis; Supervision; Funding acquisition; Validation; Investigation; Methodology; Writing—original draft; Writing—review and editing. **Andrea Senff-Ribeiro**: Formal analysis; Validation; Investigation; Visualization; Methodology. **Teele Karafin**: Formal analysis; Validation; Investigation; Visualization; Methodology. **Alexandra Lespagnol**: Formal analysis; Validation; Investigation; Visualization; Methodology. **Joane Honoré**: Validation; Investigation; Methodology. **Virginie Baylot**: Formal analysis; Validation; Investigation; Visualization; Methodology. **Josette Banroques**: Conceptualization; Formal analysis; Validation; Investigation; Visualization; Methodology. **N Kyle Tanner**: Conceptualization; Formal analysis; Supervision; Validation; Investigation; Visualization; Methodology; Writing—original draft. **Nathalie Chamond**: Resources. **Jordan D Dimitrov**: Conceptualization; Formal analysis; Validation; Investigation; Visualization; Methodology; Writing—original draft. **Johan Hoebeke**: Conceptualization; Formal analysis; Supervision; Validation; Investigation; Visualization; Methodology; Writing—original draft. **Nathalie M Droin**: Conceptualization; Validation; Investigation; Visualization; Methodology. **Bastien Job**: Formal analysis; Validation; Visualization. **Jonathan Piard**: Formal analysis; Validation; Investigation; Visualization; Methodology. **Ulrich-Axel Bommer**: Conceptualization; Validation; Visualization; Writing—original draft; Writing—review and editing. **Kwang-Wook Choi**: Conceptualization; Validation; Visualization; Writing—original draft; Writing—review and editing. **Sara Abdelfatah**: Formal analysis; Validation; Investigation; Visualization;

Methodology. **Thomas Efferth**: Conceptualization; Formal analysis; Validation; Investigation; Visualization; Methodology; Writing—original draft. **Stephanie B Telerman**: Conceptualization; Data curation; Formal analysis; Supervision; Validation; Investigation; Visualization; Methodology; Writing—original draft; Writing—review and editing. **Felipe Correa Geyer**: Formal analysis; Validation; Methodology. **Jorge Reis-Filho**: Formal analysis; Supervision; Validation; Investigation; Visualization; Methodology. **Adam Telerman**: Conceptualization; Resources; Data curation; Formal analysis; Supervision; Funding acquisition; Validation; Investigation; Visualization; Methodology; Writing—original draft; Project administration; Writing—review and editing.

## Disclosure and competing interests statement
The authors declare no competing interests.

# Expanded View Figures

**Figure EV1.   Effect of sEVs on apoptosis measured by TUNEL.**

(**A**) TUNEL assay on reporter cells alone (*WT* thymocytes). (**B**) Reporter thymocytes co-cultured with sEVs derived from γ-irradiated *WT* thymocytes. (**C**) Reporter thymocytes co-cultured with sEVs derived from γ-irradiated *Tctp⁻/⁺* thymocytes. (**D**) Relative percentage of TUNEL positive cells. Mean ± SEM (independent biological replicates $n = 4$ conditions) (ANOVA ***$P < 0.001$). Data information: (**A–D**) Scale bars: 400 µm. Source data are available online for this figure.

▶

## A  Reporter thymocytes without addition of sEVs

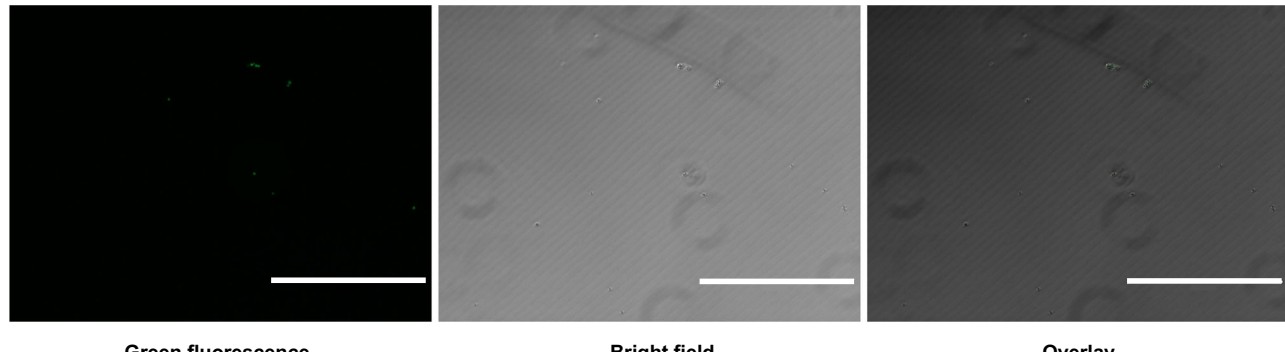

| Green fluorescence | Bright field | Overlay |

## B  Reporter thymocytes supplemented with sEVs from γ-irradiated *WT* mice

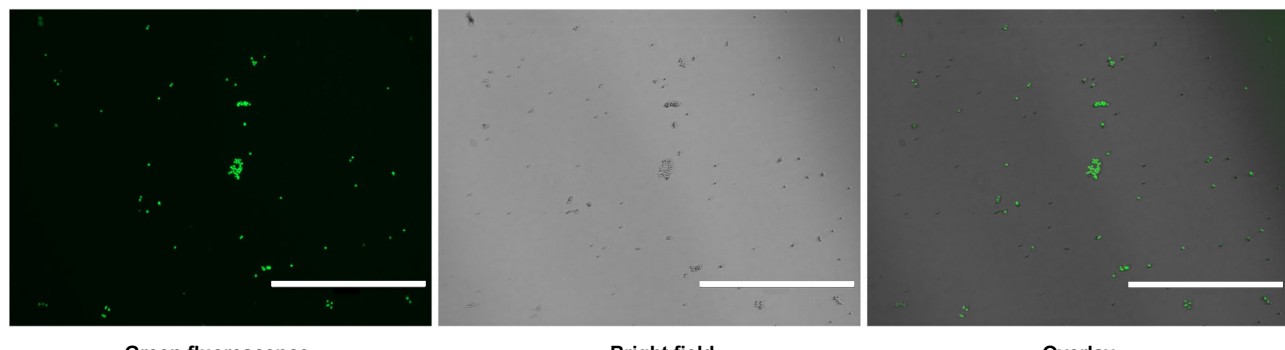

| Green fluorescence | Bright field | Overlay |

## C  Reporter thymocytes supplemented with sEVs from γ-irradiated *Tctp⁻ᐟᶠ⁻* mice

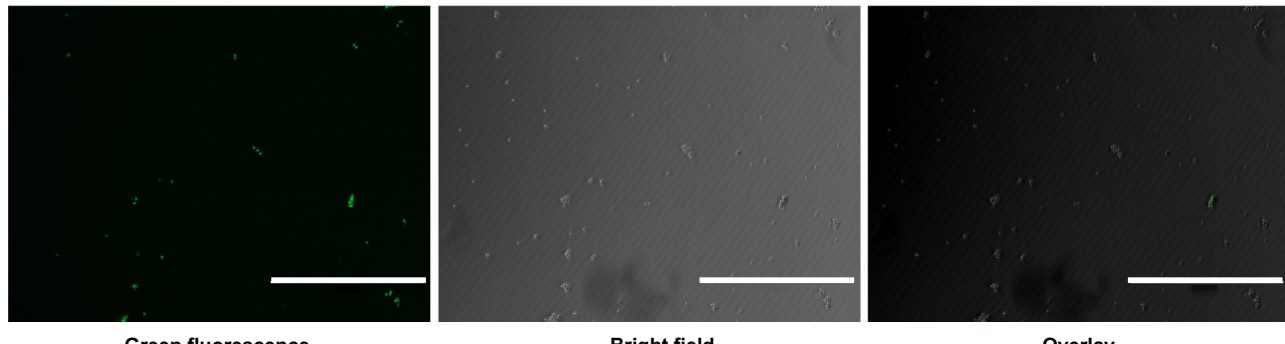

| Green fluorescence | Bright field | Overlay |

## D

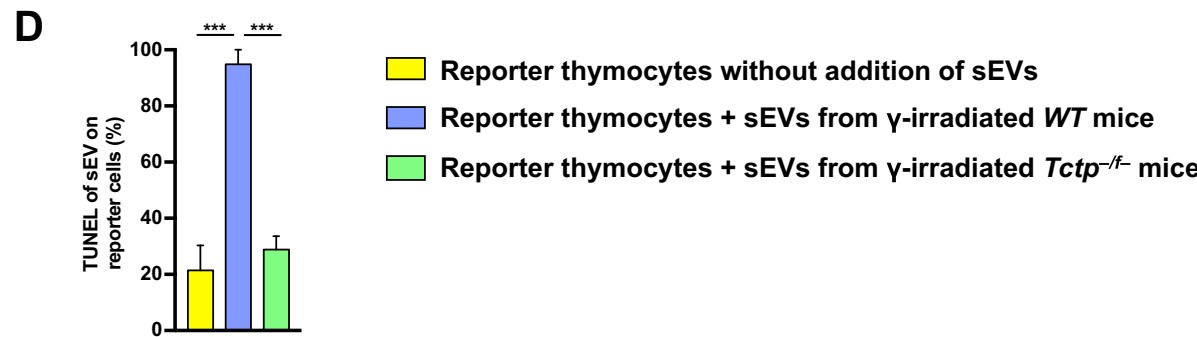

■ Reporter thymocytes without addition of sEVs
■ Reporter thymocytes + sEVs from γ-irradiated *WT* mice
■ Reporter thymocytes + sEVs from γ-irradiated *Tctp⁻ᐟᶠ⁻* mice

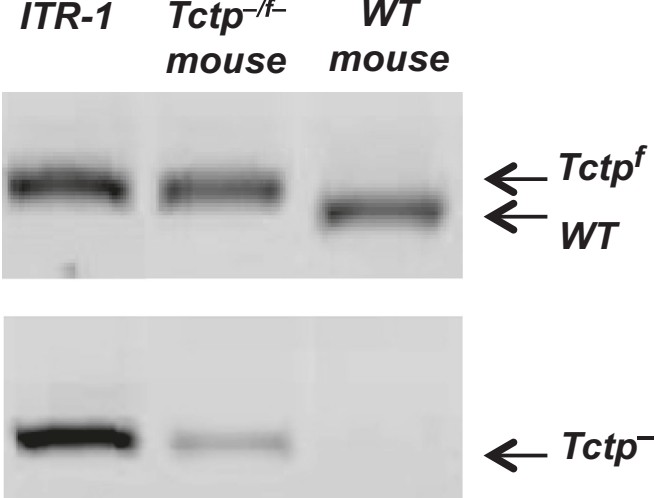

**Figure EV2.  *Tctp* genotyping of ITR-1 cells.**

PCR signals of ITR-1 cells bearing one conditional allele (*Tctp ᶠ*) and one constitutive knockout allele (*Tctp ⁻*).

**A**

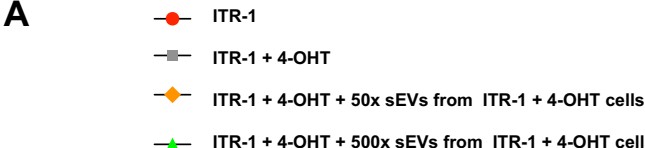

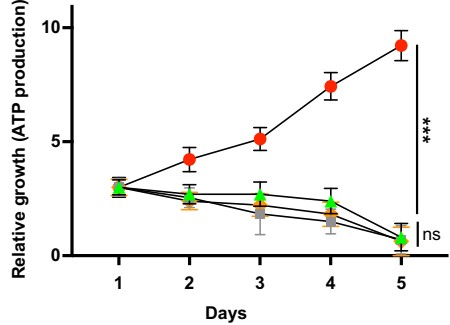

**B**

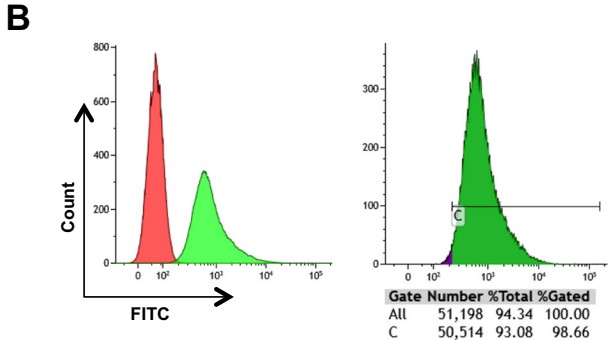

**C**

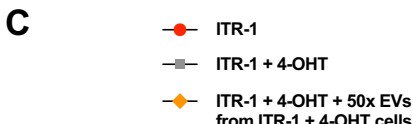

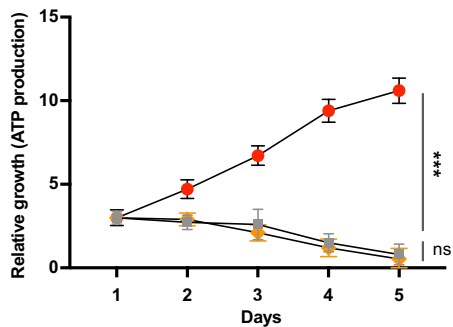

**Figure EV3.   Complementation experiments: effect of sEVs and EVs from 4-OHT treated cells.**

(A) Effect of sEVs from 4-OHT treated cells on the growth of 4-OHT treated cells. ITR-1 cells (red), ITR-1 cells treated with 4-OHT (gray). 4-OHT treated ITR-1 cells complemented with two concentrations of sEVs originating from 4-OHT treated ITR-1 cells ($17 \times 10^7$ sEVs were added to 20,000 cells, 50x excess) (orange) or $170 \times 10^7$ sEVs, 500x excess (green) (independent biological replicates $n = 3$). (B) Representative experiment showing the uptake of FITC labeled sEVs derived from 4-OHT treated ITR-1 cells by 4-OHT treated ITR-1 cells. Left graph: Unlabeled 4-OHT treated ITR-1 cells alone (red), uptake of FITC labeled sEVs from 4-OHT treated ITR-1 cells by 4-OHT treated ITR-1 cells (green). Right graph: FITC negative population (purple), FITC positive gated population (green) (C gate). The value of the gating is displayed below the graph. (C) Effect of EVs (sEVs + NVs) from 4-OHT treated cells on the growth of 4-OHT treated cells. ITR-1 cells (red), ITR-1 cells treated with 4-OHT (gray). 4-OHT treated ITR-1 cells complemented with 50x excess of EVs originating from EVs from 4-OHT treated ITR-1 cells (orange) ($17 \times 10^7$ EVs were added to 20,000 cells) (independent biological replicates $n = 4$). Data information: Statistical analysis was performed using Mann–Whitney test (A, C) Mean ± SEM. ns (not significant) ***$P < 0.001$. This control in vitro complementation experiment aims to assess the effect of sEVs, derived from 4-OHT treated ITR-1 cells, on cell growth. The conclusion of this experiment is that even a 500x excess of sEVs derived from 4-OHT treated ITR-1 cells is unable to restore tumor cell growth despite a highly efficient uptake of these sEVs. Source data are available online for this figure.

     

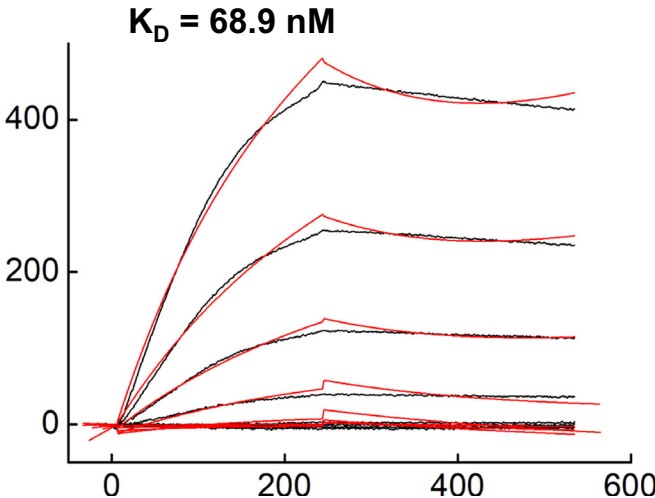

**Figure EV4.  Real-time interaction analysis of the binding of DDX3 to immobilized TCTP.**

Real-time interaction profiles of the binding of DDX3 to immobilized TCTP. The experimental data are presented in black and the global analysis fit are in red. The concentration of DDX3 ranged from 100 nM to 0.171 nM. The measurements were performed at 25 °C. Source data are available online for this figure.

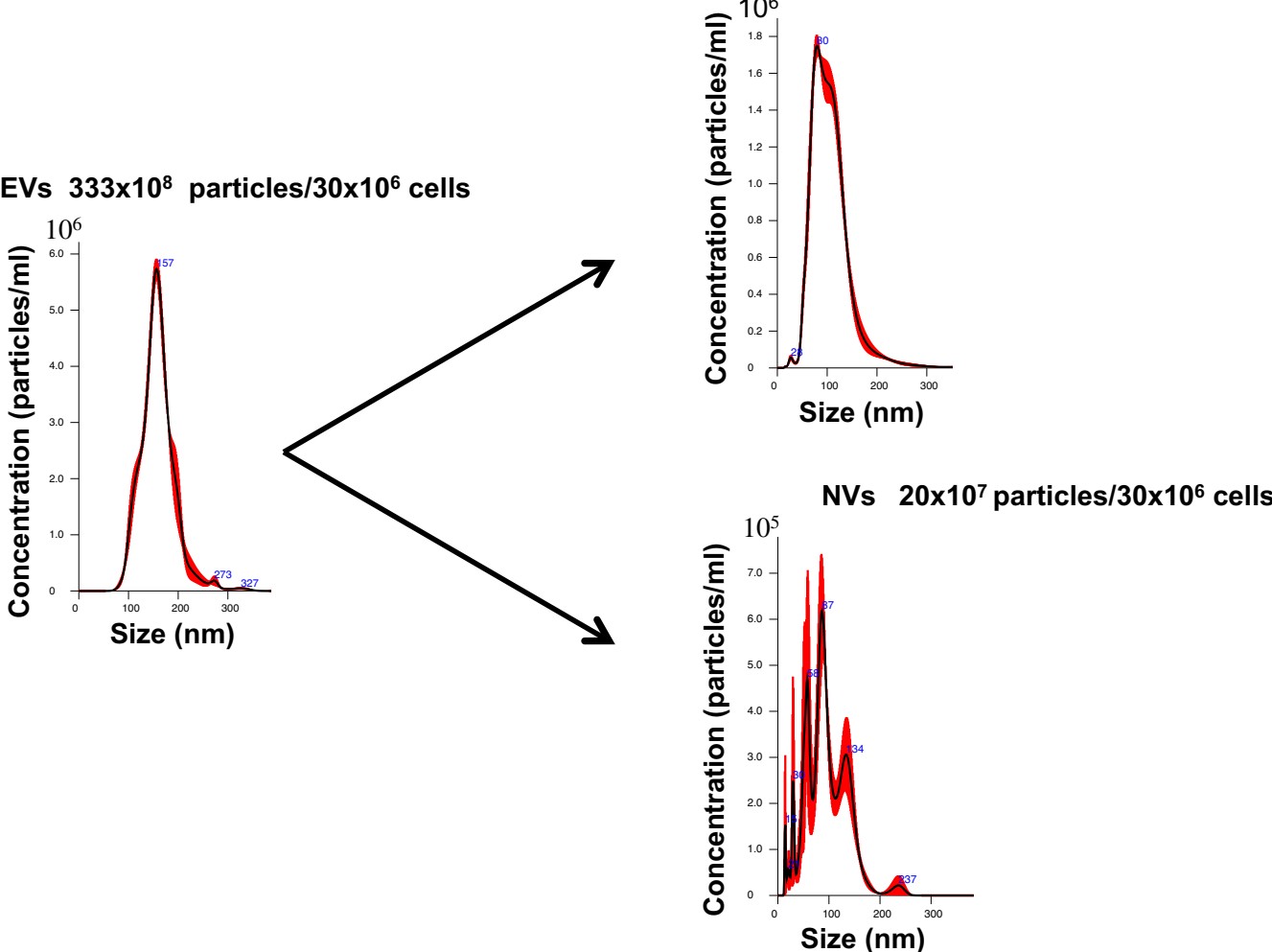

**Figure EV5.  Evaluation of sEV yield following high-resolution iodixanol gradient centrifugation.**

The counts of EVs produced by $30 \times 10^6$ thymocytes (mean $= 253 \times 10^8$) (independent biological replicates $n = 4$) compared to the counts of sEVs (mean $= 72 \times 10^7$) (independent biological replicates $n = 8$) and NVs ($16 \times 10^7$ mean) (independent biological replicates $n = 6$) indicate a loss of 57.5 fold {$(253 \times 10^8/(72\ 10^7 + 16 \times 10^7)$)} of sEVs together with NVs. Each sEV sample was diluted for an initial count to comply with the precision of the Nanosight measurements ranging from $10^6$ to $10^{10}$, being the most reliable window. Hence all the values presented take into account these dilutions. This 28.75 fold "loss" or more accurately "cleaning" using high-resolution iodixanol centrifugation remains so far one of the most precise methods to obtain purified sEVs. An entire body of literature shows that repeated steps of ultracentrifugation would also partially damage the exosomes and result in an important loss of particles. For these reasons, in the experiments using sEVs in vitro and in vivo, we used an excess of 50x of sEVs whether derived from thymocytes or cell lines.

