## [Peer Review File · EMBO Reports]

TCTP regulates genotoxic stress and tumorigenicity via intercellular vesicular signaling

Adam Telerman, Robert Amson, Andrea Senff-Ribeiro, Teele Karafin, Alexandra Lespagnol, Joane Honoré, Virginie Baylot, Josette Banroques, N. Kyle Tanner, nathalie chamond, Jordan Dimitrov, Johan Hoebeke, Nathalie DROIN, Bastien Job, Jonathan Piard, Ulrich-Axel Bommer, Kwang-Wook Choi, Sara Abdelfatah, Thomas Efferth, Stephanie Telerman, Felipe Correa Geyer, and Jorge Reis-Filho

Corresponding author(s): Adam Telerman (atelerman@gmail.com)

Review Timeline:

Submission Date:	22nd Sep 23
Editorial Decision:	17th Oct 23
Revision Received:	29th Nov 23
Editorial Decision:	18th Jan 24
Revision Received:	4th Feb 24
Accepted:	21st Feb 24

Editor: Achim Breiling

Transaction Report:

Dear Prof. Telerman,

Thank you for the submission of your research manuscript to EMBO reports. I have now received the reports from the three referees that were asked to evaluate your study, which can be found at the end of this email.

As you will see, the referees think that the findings are of high interest. However, referees #2 and #3 have several comments, concerns, and suggestions, indicating that a major revision of the manuscript is necessary to allow publication of the study in EMBO reports. As the reports are below, and all the referee concerns need to be addressed, I will not detail them here.

Given the constructive referee comments, I would like to invite you to revise your manuscript with the understanding that all referee concerns must be addressed in the revised manuscript and in a detailed point-by-point response. Acceptance of your manuscript will depend on a positive outcome of a second round of review. It is EMBO reports policy to allow a single round of revision only and acceptance of the manuscript will therefore depend on the completeness of your responses included in the next, final version of the manuscript.

- 1) a .docx formatted version of the final manuscript text (including legends for main figures, EV figures and tables), but without the figures included. Figure legends should be compiled at the end of the manuscript text.
- 2) individual production quality figure files as .eps, .tif, .jpg (one file per figure), of main figures (up to 8) and EV figures. Please upload these as separate, individual files upon re-submission.

- 4) a complete author checklist, which you can download from our author guidelines (<https://www.embopress.org/page/journal/14693178/authorguide>). Please insert page numbers in the checklist to indicate where the requested information can be found in the manuscript. The completed author checklist will also be part of the RPF.

- 5) that primary datasets produced in this study (e.g. RNA-seq, ChIP-seq, structural and array data) are deposited in an

appropriate public database. If no primary datasets have been deposited, please also state this in a dedicated section (e.g. 'No primary datasets have been generated and deposited'), see below.

The accession numbers and database should be listed in a formal "Data Availability" section (placed after Materials & Methods) that follows the model below. This is now mandatory (like the COI statement). Please note that the Data Availability Section is restricted to new primary data that are part of this study. This section is mandatory. As indicated above, if no primary datasets have been deposited, please state this in this section

Data availability

8) Regarding data quantification and statistics, please make sure that the number "n" for how many independent experiments were performed, their nature (biological versus technical replicates), the bars and error bars (e.g. SEM, SD) and the test used to calculate p-values is indicated in the respective figure legends (also for potential EV figures and all those in the final Appendix). Please also check that all the p-values are explained in the legend, and that these fit to those shown in the figure. Please provide statistical testing where applicable. Please avoid the phrase 'independent experiment', but clearly state if these were biological or technical replicates. Please also indicate (e.g. with n.s.) if testing was performed, but the differences are not significant. In case n=2, please show the data as separate datapoints without error bars and statistics. See also: <http://www.embopress.org/page/journal/14693178/authorguide#statisticalanalysis>

9) Please add scale bars of similar style and thickness to all the microscopic images, using clearly visible black or white bars (depending on the background). Please place these in the lower right corner of the images themselves. Please do not write on or near the bars in the image but define the size in the respective figure legend.

10) Please also note our reference format:

12) We now use CRedit to specify the contributions of each author in the journal submission system. CRedit replaces the author contribution section. Please use the free text box to provide more detailed descriptions and do not provide your final manuscript text file with an author contributions section. See also our guide to authors: <https://www.embopress.org/page/journal/14693178/authorguide#authorshipguidelines>

13) We would encourage you to use 'Structured Methods', our new Materials and Methods format. According to this format, the

Materials and Methods section should include a Reagents and Tools Table (listing key reagents, experimental models, software and relevant equipment and including their sources and relevant identifiers) followed by a Methods and Protocols section in which we encourage the authors to describe their methods using a step-by-step protocol format with bullet points, to facilitate the adoption of the methodologies across labs. More information on how to adhere to this format as well as downloadable templates (.doc or .xls) for the Reagents and Tools Table can be found in our author guidelines (section 'Structured Methods'):

14) Please add 5 keywords to the manuscript text file and order the manuscript sections like this, using these names: Title page - Abstract - Keywords - Introduction - Results - Discussion - Materials and Methods - Data availability section - Acknowledgements - Disclosure and Competing Interests Statement - References - Figure legends - Expanded View Figure legends

Finally, please note that all corresponding authors are required to supply an ORCID ID for their name upon submission of a revised manuscript. Please find instructions on how to link the ORCID ID to the account in our manuscript tracking system in our Author guidelines: <http://www.embopress.org/page/journal/14693178/authorguide#authorshipguidelines>

I look forward to seeing a revised version of your manuscript when it is ready. Please let me know if you have questions or comments regarding the revision.

Yours sincerely,

Referee #1:

I have read with great interest the ms by Amson et al. "TCTP regulates intercellular vesicular signaling in genotoxic stress and tumorigenicity". It is an important and impressive piece of work substantially elucidating the interplay between TCTP and p53 in tumorigenesis. The authors have previously shown the direct mutual negative regulation between these two proteins, which was an important breakthrough in the field of cancerogenesis. Here, they demonstrate that TCTP is required for functional sEVs, the so-called exosomes, which pave the avenue for studying the long distance effects of TCTP in regulation of apoptosis. Without any doubt, this excellent work is novel, of great scientific importance and for this reasons it meets all requirements for publishing in EMBO Reports.

I will first comment on certain technical aspects because they seem to me especially important. The authors provide a genetic model that was unavailable so far - namely, Tctp conditional knockout mice. The studies that follow were impossible to investigate without using these mice. The full TCTP knockout mice are lethal at embryonic day E=7.5. The authors also used high-resolution iodixanol density gradient centrifugation to isolate the sEVs. This is a very precise technique, but extremely heavy to carry out. This is why most of the studies in the past have used the 120,000g pellet to isolate the exosomes with the problem of having a mixed population of sEVs and non-vesicular structures. For this reason, using the high-resolution iodixanol density gradient centrifugation constitutes a major mean to carry out this study.

The authors ask two fundamental questions:

1. What is the difference between induced and spontaneous sEVs secretion in cells with and without TCTP?
2. How cells exposed to γ -irradiation signal apoptotic commands to cells that have not been irradiated?

They clearly demonstrate that:

1. TCTP is required for sEVs signaling of cell death following irradiation. Indeed the sEVs from the conditional knockout mice have no effect on reporter cells.
2. When they cross p53 null mice, which are prone to spontaneous tumor formation, with the conditional Tctp knockout mice they observe a significantly prolonged survival when Tctp is ablated. This is by its own a major finding that warrants publication of this paper in a journal of the range of EMBO Reports.
3. They derived an inducible Tctp knockout cell line (ITR cells), which was instrumental in proving that Tctp is required for oncogenic signaling by sEVs.
4. Ultimately they provide a convincing mechanism for the recruitment of RNAs by the TCTP-DDX3 complex.
5. Last but not least: when p53 knockout mice are treated with daily intake of sertraline, which inhibits the function of TCTP, these p53 KO mice have a significantly prolonged survival.

In conclusion: this article provides entirely novel data on apoptosis and oncogenic signaling by sEVs, and is clearly of high value for the scientific community in the field of cancerology. It is well written and very clear. I suggest that it would be published as it is in the present form. I did not find unclear or doubtful elements in the manuscript, which is rare.

Minor point for eventual addition:

1. It would be interesting for the reader to have access to the full size Western blots. Especially those that document the fractionation of the subpopulations of the vesicles. They may be included in the supplementary material. Nowadays, this requirement is demanded by numerous journals.

Referee #2:

In the article 'TCTP regulates intracellular vesicular signalling in genotoxic stress and tumorigenicity' the authors have established a new mouse model of TCTP conditional knockout to validate previous findings and discover how TCTP generated EVs contribute to tumorigenesis. The survival data of the TCTP knockout mice is a strong selling point of this manuscript, but it is not convincingly proven that increased survival is due to a decrease in EVs reaching the tumour tissue. Figure 5 about the involvement of RNA is weak and needs several additions to validate the mechanism proposed at the end.

Major comments:

1. The weakest point of this manuscript is that it is unclear whether any of the mechanisms for how TCTP is working are demonstrated to underlie the phenotype seen in mice in figure 3B. Do the authors actually see a decreased amount of EVs in the serum of these mice? Which cells in the mouse would be responsible for EV production? Just the tumour cells or any other cells? Thymocytes seem to also produce EVs when TCTP is present and figure 1A suggest a whole body inducible knockout is used.
2. At best the authors establish cell lines from the tumours and isolate EVs from there to incubate cells that are then xenografted in mice in Figure 4J. Can tumour growth also be restored using vesicles from thymocytes used in Figure 1? And a control of using EVs from Tamoxifen induced ITR cells is needed to prove that it is RNA content that is the crucial restorative factor as figure 5 suggests.
3. Figure 1 is all performed in the presence of radiation. Is radiation necessary to see p53? It is not entirely clear what this figure contributes to the manuscript as none of the following results use radiation. Are these effects miRNA/ DDX3 dependent?
4. Sirois et al., (2013) demonstrated that Caspase-3 facilitated the export of TCTP and thereby a bystander anti-apoptotic effect. So the finding here that TCTP can also propagate pro-apoptotic signals unveils a significant context-dependency of TCTP's activity. This needs addressing or at least discussion.
5. For all the FACS analyses, it is a bit unclear what is what. sEV red - is this control non-stained EVs on their own, or sEVs given to cells? The FITC labelled population is overlapping the non-FITC labelled population by a lot. (but not in figure 2). If these are just the EVs, how are the authors certain the graph on the right is actually showing FITC-labeled vesicles in cells? It is not explained what overlay is and why the C-gate is started where it is.
6. The vesicle isolation suggest TCTP is expressed in the vesicles. What happens when the vesicles are imported, do the cells that takes these up use this TCTP? Can an increased TCTP be seen in Figure 4M, similar to 4K. Would be good to show TCTP levels here.
7. It is unclear to what extent miRNA loading in the vesicles is actually contributing to tumour formation. Do vesicles of TCTP KO cells not promote tumour formation?
8. Why did the authors study DDX3? In a previous publication PMID 23122550 the authors suggest MSS4 could bind. Why chose DDX3 or Ded1 as target? To what extent does loss of DDX3 simulate loss of TCTP?
9. The data that the drugs bind TCTP especially with previous publications by the authors is convincing, but are they actually preventing EV formation? And loading of protein/ RNA in vesicles. This needs to be shown. Figure 5L is not convincing. What is the really high band and why is no band visible anywhere in the thioridazine lane. Did the cells die?
10. Some validation of the data in Figure 5. Can these microRNAs be found in tumours. Is it the RNA content of these vesicles that prolongs life? Is that DDX3 dependent? Other RNA binding proteins?

Minor comments:

1. The following sentence needs a reference or if not yet published, these data need to be shown: 'We previously found that cofactors enhance the RNA-dependent ATPase activity of Ded1 presumably by acting like chaperones (they are not chaperones though, by current definitions) to facilitate the folding of the protein into an active conformation'
2. In Figure 2 several cell lines and several shRNAs for TCTP are used. Unless mistaken, only one SH7239 is used in following experiments. Is this info needed?
3. 2H It is unclear how this was measured. Was the amount of FITC sEV set to 100% in sEV? Raw values in supp would be useful here to see if only one cell had taken it up or whether it is at least a substantial proportion of cells.
4. A more comprehensive introduction with what is known already based on previous papers of the authors as well as literature in the field is needed.
5. In reference to Figure 2, authors say effect of TCTP kd is independent of p53 status. This is a bit of an overstatement as this is done in different cell lines. The authors can say it is likely not p53 dependent, but to state it like this a p53 KO or KD is needed.
6. The discussion regarding the false negatives in figure 5 is too long and distracts from the paper. Not sure why this is needed here. Could go to supplemental data and more briefly discussed in the main text, with possible elution to this in the supplemental text.

Referee #3:

This manuscript focuses on TCTP as an antagonist role of p53 during tumorigenesis, focusing on its effects on small EV biogenesis and intercellular communication in vivo. The authors generate a compound knockout mouse model lacking p53 combined with the inducible deletion of TCTP. The concomitant loss of TCTP and p53 results in improved overall survival and a change in the tumor spectrum such that an increased number of sarcomas are observed in this animal. The authors demonstrate that both cells derived from TCTP deficient animals have reduced numbers of sEVs, implicating this protein in EV biogenesis. In addition, TCTP does not directly bind RNA; rather, it was found to bind to DDX3, which serves as an RNA-binding protein for incorporation of certain small RNAs, such as let-7c-5p into sEVs. Chemicals that disrupt the interaction of TCTP to DDX3 are able to phenocopy the effects on mouse survival in p53 deficient animals that were observed upon combined p53 and TCTP genetic loss in vivo; hence, the authors propose that TCTP is critical for producing sEVs containing small RNAs that mediate the pro-tumorigenic effects of p53 loss. Although the results support a role for TCTP in modulating p53 function in vivo, the precise effects of TCTP-induced sEVs on these tumor-promoting phenotypes remain murky. Overall, the results lack mechanistic depth and do not fully support the conclusions that have been drawn.

- 1) The in vivo results in Fig 3 are confusing with regard to the role of sEVs. The authors demonstrate a reduction in lymphoma as well as reduced sEVs produced by thymocytes. However, a broad spectrum of sarcomas does develop in p53/TCTP double deficient mice suggesting that sarcomas can develop independently of TCTP function in vivo. However, TCTP deletion ex vivo in ITR cells, a sarcoma derived from these mice, also results in reduced sEV production. In a transplant model, sEVs are able to rescue the effects of TCTP loss in ITR cells. Are sEVs from required for the maintenance of TCTP-deficient ITR cells but not for their initiation? If so, further clarification of the role of TCTP-derived sEVs in tumor initiation vs. maintenance is needed. As it stands, the results are more confusing than illuminating, especially with regard to the role of sEVs in tumor promotion.
- 2) The authors propose that TCTP promotes that sEV-mediated transfer of oncogenic signals, namely small RNAs, from one tumor cell to another. However, it is unclear from the results whether the sEVs rescue experiments, which employ a supraphysiological amount of sEVs (approximately 10⁸), involve the direct transfer of specific oncogenic signals or RNA species among tumor cells. sEVs may have broader non-RNA-dependent effects on both tumor and non-tumor cells which have not been considered. Overall, the results lack mechanistic depth and do not support the conclusions that have been drawn.
- 3) Details regarding the Cre driver are needed to interpret the in vivo model. What is the transgenic driver of Cre expression. Does the promoter effect explain why the deletion of TCTP is incomplete, not only in vivo but also when cells are treated ex vivo (e.g. Fig 4A)?
- 4) In Fig 2, the key studies in breast cancer cell lines have been performed with a single shRNA targeting TCTP. These should be reproduced with a second shRNA to control for off-target effects. Moreover, the growth assays in Fig 2I should include sEV rescues from from TCTP knockdown cells to ascertain whether the effect of TCTP-deficiency is lack of sEV production or due to lack of cargo transfer (as Fig 5 later implies).
- 5) The in vivo studies using sertraline in Fig 5 are low resolution in scope and do not convincingly establish a functional role for TCTP-induced sEVs in mediating the observed effects on survival. Is the tumor spectrum similar to that observed with the double knockout mice in Fig 3. What is the effect of sertraline on sEV production from p53 deficient thymocytes?

Rebuttal letter:

Referee #1:

I have read with great interest the ms by Amson et al. "TCTP regulates intercellular vesicular signaling in genotoxic stress and tumorigenicity". It is an important and impressive piece of work substantially elucidating the interplay between TCTP and p53 in tumorigenesis. The authors have previously shown the direct mutual negative regulation between these two proteins, which was an important breakthrough in the field of cancerogenesis. Here, they demonstrate that TCTP is required for functional sEVs, the so-called exosomes, which pave the avenue for studying the long distance effects of TCTP in regulation of apoptosis. Without any doubt, this excellent work is novel, of great scientific importance and for this reasons it meets all requirements for publishing in EMBO Reports.

I will first comment on certain technical aspects because they seem to me especially important. The authors provide a genetic model that was unavailable so far - namely, Tctp conditional knockout mice. The studies that follow were impossible to investigate without using these mice. The full TCTP knockout mice are lethal at embryonic day E=7.5. The authors also used high-resolution iodixanol density gradient centrifugation to isolate the sEVs. This is a very precise technique, but extremely heavy to carry out. This is why most of the studies in the past have used the 120,000g z

Minor point for eventual addition:

1. It would be interesting for the reader to have access to the full size Western blots. Especially those that document the fractionation of the subpopulations of the vesicles. They may be included in the supplementary material. Nowadays, this requirement is demanded by numerous journals.

We thank the referee for his encouraging comments. We will include the Western blots in the specific section.

Referee #2:

In the article 'TCTP regulates intracellular vesicular signalling in genotoxic stress and tumorigenicity' the authors have established a new mouse model of TCTP conditional knockout to validate previous findings and discover how TCTP generated EVs contribute to tumorigenesis. The survival data of the TCTP knockout mice is a strong selling point of this manuscript, but it is not convincingly proven that increased survival is due to a decrease in EVs reaching the tumour tissue. Figure 5 about the involvement of RNA is weak and needs several additions to validate the mechanism proposed at the end.

We thank referee 2 for his constructive criticism.

Major comments:

1. The weakest point of this manuscript is that it is unclear whether any of the mechanisms for how TCTP is working are demonstrated to underlie the phenotype seen in mice in figure 3B. Do the authors actually see a decreased amount of EVs in the serum of these mice?

We understand the concern of the referee that we don't show directly in our TCTP conditional knockout mice that there is a drop in the number of EVs. However, in the present work, we focused on small extracellular vesicles (sEVs). Indeed, as we worked on sEVs, it was necessary for our study to have an additional step of *in vitro* culture, both for the thymocytes from the irradiated TCTP conditional knockout (*Tctp*^{-f/-}) mice and for the tumors in the *Trp53*^{-/-};*Tctp*^{-f/-} mice presented in Figure 3. To address this concern in the survival curve of Figure 3, we would need a direct quantification of sEVs in the serum, which would require harvesting the serum of hundreds of mice at each time point in order to have enough sEVs to provide reliable quantification. In addition, as we can't predict in advance when a mouse will form its tumor, we would not be able to coordinate their serum harvesting, and it would probably not be possible to provide reproducible results for sEVs.

Referee 2 suggests a quantification of EVs in the serum of mice: It would have been feasible to provide some numbers using semi-quantitative EV analysis in the serum of mice. The problem is that these EV analyses would represent a mixed population of vesicles. This kind of data is not acceptable anymore for publication in any highly rigorous journal, as this has been shown to quantify EVs but not specifically sEVs. For sEVs, this requires purification by high-resolution iodixanol density gradient centrifugation, which is not possible at the scale of mice serum. In the present study, the focus was specifically on the characterization of small extracellular vesicles (sEVs), which are the ones containing TCTP.

EVs (extracellular vesicles) are a mixture of both sEVs (small extracellular vesicles) and NV (non-vesicular structures), which are different, as documented with great care (Jeppesen *et al*, 2019). Indeed, this field has completely changed after this publication: "**Jeppesen DK, Fenix AM, Franklin JL, Zhang Q, Zimmerman LJ, Liebler DC, Ping J, Liu Q, Evans R et al (2019) Reassessment of Exosome Composition. Cell 177: 428-445**".

Before that publication, the established methodology for the isolation of exosomes was quite fast and straightforward. The problem is that many studies published on exosomes, except when supplementary purification steps were used (such as sucrose gradient centrifugation), have proven to lack accuracy on vesicle subtypes and typically included a mixture of vesicles.

For example, here is the recent reevaluation of papers on the mechanism of miRNAs uptake through proteins, which shows that these proteins are actually not present in small extracellular vesicles. Comment of Coffey's group in Cell 2019: "*None of the frequently reported exosomal RBPs (Hagiwara et al, 2015; Mateescu et al, 2017) that we investigated (Ago1-4, annexin A2, RPS3, RPS8, EEF2, EEF1A1, MVP, PARK7/DJ1, hnRNPA2B1, and GAPDH) were associated with classical CD63-, CD81-, or CD9-positive exosomes*" (Jeppesen *et al.*, 2019).

Among the papers reevaluated are: (Hagiwara *et al.*, 2015; Villarroja-Beltri *et al*, 2013).

This is why the best that one could offer with nowadays technology is an *ex-vivo* proxy consisting of a tumor cell line derived from a tumor of these double knockouts, and we were very lucky to rescue such an inducible TCTP knockout cell line (ITR-1 cells). The approach is similar to studies in the field of stem cell biology, where one needs, in most circumstances, to use a proxy with an *ex-vivo* culture system (Amson *et al*, 2012).

We completely agree to tone down our conclusions and point specifically to the fact that we don't have a direct evidence of the drop in sEVs in the conditional TCTP knockout mice and say that we used an *ex-vivo* proxy to document the decrease of sEVs.

We have been more specific in our discussion of the data showing that tumor formation after deletion of TCTP led to a decreased secretion of sEVs, stating that this is only a partial explanation and that there are probably many other factors but that these are not the focus of the present paper.

The following has been added to the manuscript:

In the introduction:

“Extracellular vesicles (EVs) consist of a heterogeneous population (small extracellular vesicles (sEVs) and non-vesicular structures (NVs)). Many studies in the past and still now are performed on these mixed populations because the method to isolate these EVs is straightforward. However, recent work on the composition of EVs redefined the field, providing methodologies that increased accuracy in the isolation of sEVs from NVs using high-resolution iodixanol gradients, together with the availability of new markers (Crescitelli *et al*, 2021; Gyuris *et al*, 2019; Jeppesen *et al.*, 2019; Zhang *et al*, 2018). Given the progress made in the characterization of sEVs, we can now properly address questions about the genetic and molecular way through which TCTP defines the content and function of sEVs”.

In the results:

Section: Tctp-regulated sEV-signaling in genotoxic stress-induced apoptosis (Figure1).

“While the methodologies to isolate sEVs have improved, quantification and accurate analysis of these specific thymocytes derived vesicles in the serum of mice still requires an extra *in vitro* step. This is to avoid a mixture of sEVs derived from different organs as well as the necessity to use an excessive number of mice because of the low yield of sEVs purification”.

Section: sEV-regulated malignant transformation signaling is Tctp-dependent.

“To this aim, we developed an *ex-vivo* proxy consisting of a cell line from one of these sarcomas and called it ITR-1 (standing for Inducible Tumor Reversion cell line)”.

And “These experiments using an *ex-vivo* proxy (ITR-1 cells) indicate that sEVs transfer of oncogenic material is regulated in a Tctp-dependent manner.”

In the discussion:

“This inducible ablation of *Tctp* and its consequences on tumor formation in mice is probably due to a series of events regulated by Tctp (Bommer & Telerman, 2020). In the context of the present study, to address the question of whether Tctp in the sEVs could at least be partially responsible for this reprogramming of tumors, we had to use an *ex-vivo* proxy. Hence, the complementation experiments using the inducible *Tctp*-knockout cell line ITR-1, showed that these *Trp53*^{-/-} cells that lost their malignancy as a result of deleting both *Tctp* alleles, regained it upon supplementation with sEVs from the malignant parental ITR-1 cell line (Fig 4). These results provide evidence that *Trp53*^{-/-} driven tumorigenesis, *ex-vivo*, is reprogrammable/reversible by ablation of the Tctp-sEVs circuitry”.

As well as:

“On the contrary, the spontaneous secretion of sEVs by cancer cells, as examined in the present work, is not likely p53 dependent, whether wild type p53, loss or gain of function mutants. This led us to the conclusion that TCTP in sEVs signaling is required and determines the fate of recipient cells regardless of whether it is apoptosis or malignant transformation”.

Which cells in the mouse would be responsible for EV production? Many cell types in the organism secrete sEVs.

Just the tumour cells or any other cells? Other cells also, this is why it would be confusing to quantify it from mice serum

Thymocytes seem to also produce EVs when TCTP is present and figure 1A suggest a whole body inducible knockout is used.

Fig 1A is about our strategy and is indeed a whole body inducible knockout. As mentioned now in the section Materials and Methods: The Cre-Ert2 gene (Tamoxifen dependent) is inserted into the Rosa 26 locus and under the control of the Rosa 26 promotor.

Fig 1B is not about sEVs, we just measure in organs the amount of TCTP in WT mice versus those deleted in TCTP.

2. At best the authors establish cell lines from the tumours and isolate EVs from there to incubate cells that are then xenografted in mice in Figure 4J. Can tumour growth also be restored using vesicles from thymocytes used in Figure 1?

The thymocytes of non-irradiated mice produce a very low level of sEVs. Upon irradiation, WT thymocytes undergo apoptosis. There was no reason to use irradiated thymocytes in our ITR-1 cancer cell line model in Fig 4.

And a control of using EVs from Tamoxifen induced ITR cells is needed to prove that it is RNA content that is the crucial restorative factor as figure 5 suggests.

We addressed this question in Figure EV3: Effect of sEVs from 4-OHT treated cells on 4-OHT treated cells. "The conclusion of this experiment is that even a 500x excess of sEVs derived from 4-OHT treated ITR-1 cells are unable to restore tumor cell growth." It was not the purpose of this paper to find out what is/are the restorative RNAs. We observe that conditional deletion of TCTP results in a decrease in both proteins and RNA in the sEVs. There are probably a handful of RNAs and proteins that will be needed to restore tumor growth, but again, the only focus of this work is to establish the regulatory role of TCTP in sEVs content and function.

3. Figure 1 is all performed in the presence of radiation. Is radiation necessary to see p53?

Indeed, irradiation is necessary to induce genotoxic stress, which activates the expression and function of P53. Without irradiation, p53 is barely detectable in the wild-type thymocytes.

In the thymocytes from conditional Tctp KO mice, p53 is detectable without irradiation, but its expression is much stronger after irradiation.

It is not entirely clear what this figure contributes to the manuscript as none of the following results use radiation.

We thought that it was more logical to start out analysis on normal cells rather than directly on tumor cells. As such, we first characterized the conditional Tctp KO, and then we investigated the role played by TCTP in sEVs. Since thymocytes secrete extremely low levels of sEVs at homeostasis, a stimulus such as genotoxic stress was needed to induce sEVs secretion. We therefore analyzed sEVs secretion and content after irradiation/genotoxic stress. Indeed, this stimulus causes p53-dependent apoptosis. We then investigated whether sEVs are able to signal cell death to other cells; this is described as the Bystander effect (Azzam *et al*, 1998; Azzam & Little, 2004).

Are these effects miRNA/ DDX3 dependent?

Yes, please see Fig 5

4. Sirois et al., (2013) demonstrated that Caspase-3 facilitated the export of TCTP and thereby a bystander anti-apoptotic effect. So the finding here that TCTP can also propagate pro-apoptotic signals unveils a significant context-dependency of TCTP's activity. This needs addressing or at least discussion.

We thank reviewer 2 for drawing our attention to the work of Sirois et al., (Sirois *et al*, 2011). This work identifies activated caspase-3 as a novel regulator of TCTP export via nanovesicles, contributing hereby to an anti-apoptotic paracrine pathway. Yes, this will be addressed in the discussion. As demonstrated in our work here, this function of TCTP is indeed completely context-dependent since it signals cell death in the sEVs of irradiated thymocytes and, on the other side, malignant transformation.

5. For all the FACS analyses, it is a bit unclear what is what. sEV red - is this control non-stained EVs on their own, or sEVs given to cells?

The FACS data in Fig 1I represent the quantification of the uptake of FITC-labeled sEVs (either derived from γ -irradiated *WT* thymocytes (left panels) or *Tctp*^{-/-} thymocytes (right panels)) by reporter cells (which are wild type thymocytes). As stated in the figure legends, the red graphs represent the controls, which are thymocytes alone without any prior incubation with sEVs.

FACS analysis does not allow the visualization or counting of EVs and sEVs. Here, we use FACS to assess the uptake of FITC labeled sEVs by cells. To visualize, measure and count sEVs, another technology is needed: Nanoparticle Tracking Analysis (NTA). No need to label sEVs for NTA. This is displayed in Fig 1E,F; Fig 2C,D and Fig 4C,D.

The FITC labelled population is overlapping the non-FITC labelled population by a lot. (but not in figure 2). If these are just the EVs, how are the authors certain the graph on the right is actually showing FITC-labeled vesicles in cells? It is not explained what overlay is and why the C-gate is started where it is.

For each genotype: on the left, the green graph corresponds to thymocytes incubated in the presence of FITC labeled sEVs. On the right, the purple side of the graph represents the part of these thymocytes that did not uptake the sEVs and is therefore FITC negative: excluded from the C gating of the FITC positive population, which is in green. In these experiments, the uptake of FITC labeled sEVs was about 50%. The C gate is started after the unlabeled thymocytes, therefore FITC negative controls (red graphs).

In Figure 2, indeed, the data show that the uptake by tumor cells of FITC labeled sEVs is much higher than for primary cells (thymocytes Figure 1). This difference is common between primary cells and cancer cell lines due mainly to a difference in growth parameters. Another potential parameter to consider that could explain this difference, is their origin: thymocytes vs epithelial breast cancer cells.

The main point of these experiments was to measure whether there is a significant difference in the uptake of sEVs originating from wild type or *Tctp*^{-/-} thymocytes. Indeed, one could argue that the lack of induction of apoptosis by sEVs originating from *Tctp*^{-/-} thymocytes is due to a difference in their uptake, which is clearly not the case.

6. The vesicle isolation suggest TCTP is expressed in the vesicles. What happens when the vesicles are imported, do the cells that takes these up use this TCTP? Can an increased TCTP be seen in Figure 4M, similar to 4K. Would be good to show TCTP levels here.

Unfortunately, it is technically impossible to discriminate cellular TCTP from sEVs' TCTP in tumor cells.

7. It is unclear to what extent miRNA loading in the vesicles is actually contributing to tumour formation. Do vesicles of TCTP KO cells not promote tumour formation?

We addressed this question in Figure EV3: Effect of sEVs from 4-OHT treated cells on 4-OHT treated cells. *"The conclusion of this experiment is that even a 500x excess of sEVs derived from 4-OHT treated ITR-1 cells are unable to restore tumor cell growth."*

8. Why did the authors study DDX3?

We have chosen DDX3 because it binds TCTP (Li *et al*, 2016) and because we are familiar with Ded1-related systems binding RNA.

In a previous publication PMID 23122550 the authors suggest MSS4 could bind.

In our review paper PMID 23122550, *TPT1/ TCTP-regulated pathways in phenotypic reprogramming*, we say that: The TPT1/TCTP protein is structurally similar to MSS4. MSS4 is a guanine nucleotide exchange factor (Amson *et al*, 2013). Both TCTP and Mss4 contain structural domains of the helicase family (Takahasi *et al*, 2008).

Why chose DDX3 or Ded1 as target?

An initial collaboration with the Tanner laboratory was done because TCTP had some structural homology to LGP2, which is classified as a superfamily 2 helicase. The Tanner laboratory works with yeast Ded1, which is one of the most enzymatically active DEAD-box proteins, to determine if TCTP had any RNA-dependent activity. DDX3 is the functional human homolog (ortholog) of Ded1. Although TCTP has no detectable RNA binding affinity, we suspected that it might interact with proteins that do. Fortunately, DDX3 was previously implicated as a TCTP partner (Banroques *et al*, 2008, 2011; Li *et al.*, 2016; Senissar *et al*, 2014).

To what extent does loss of DDX3 simulate loss of TCTP?

We do not have a direct answer to this question and this is beyond the scope of this work. What we can say is that inhibiting TCTP by sertraline or thioridazine has, as a consequence, a decreased binding of DDX3 to the RNA (please see Figure 5).

9. The data that the drugs bind TCTP especially with previous publications by the authors is convincing, but are they actually preventing EV formation?

The *in vitro* data if Fig 5L were generated using purified proteins (TCTP and DDX3), Let-7c-5p and AMP-PNP. No cells are involved here.

And loading of protein/ RNA in vesicles. This needs to be shown. Figure 5L is not convincing. What is the really high band and why is no band visible anywhere in the thioridazine lane. Did the cells die?

Here again, no cells or sEVs were used. In this non-denaturing gel some Let-7c-5p may remain in the slots. We do not know why this is not visible in the last lane. Maybe because no complexes at all were generated. At any rate Let-7c-5p was not omitted because free Let-7c-5p is present in all the lanes, including the last one.

10. Some validation of the data in Figure 5. Can these microRNAs be found in tumours. Is it the RNA content of these vesicles that prolongs life? Is that DDX3 dependent? Other RNA binding proteins?

This is a relevant point that would merit a completely separate study but would require several more years; it is a different project. Not all RNA-binding proteins were tested, just DDX3. Other proteins could be implicated, but this is outside the scope of the present paper.

Minor comments:

1. The following sentence needs a reference or if not yet published, these data need to be shown: 'We previously found that cofactors enhance the RNA-dependent ATPase activity of Ded1 presumably by acting like chaperones (they are not chaperones though, by current definitions) to facilitate the folding of the protein into an active conformation'

This has been added now.

2. In Figure 2 several cell lines and several shRNAs for TCTP are used. Unless mistaken, only one SH7239 is used in following experiments. Is this info needed?

Yes, because this sh7239 was the most efficient.

3. 2H It is unclear how this was measured. Was the amount of FITC sEV set to 100% in sEV? Raw values in supp would be useful here to see if only one cell had taken it up or whether it is at least a substantial proportion of cells.

We agree that the presentation of the results was unclear, and this has now been modified. For Figure 2G, FACS experiment of MCF7 cells was taken as an example to illustrate the gating strategy for FITC-positive cells: cells that had an uptake of FITC labelled sEVs. In this example, MCF7 cells, only a proportion of cells, about 70%, have an uptake of FITC labeled sEVs. This cell line has the lowest uptake in FITC labeled sEVs. Other cell lines, such as the MDA-MB231, have a much higher sEVs uptake.

In Figure 2H, we show the quantification of the FACS experiments done for the different cell lines.

4. A more comprehensive introduction with what is known already based on previous papers of the authors as well as literature in the field is needed.

We are already citing our papers and some review papers from the field.

5. In reference to Figure 2, authors say effect of TCTP kd is independent of p53 status. This is a bit of an overstatement as this is done in different cell lines. The authors can say it is likely not p53 dependent, but to state it like this a p53 KO or KD is needed.

We completely agree and we changed it now in the results section “TCTP regulates sEV-dependent malignant growth in human breast tumor cell models”: “Therefore, this suggests that the effect of the loss of TCTP on sEV-dependent malignant growth, it is likely not p53 dependent, when analyzing the spontaneous secretion of sEVs by cancer cells”.

6. The discussion regarding the false negatives in figure 5 is too long and distracts from the paper. Not sure why this is needed here. Could go to supplemental data and more briefly discussed in the main text, with possible elution to this in the supplemental text.

The problem is that some people who look at these results may not understand the false negatives. So, in order to be cautious, we left it in the paper.

Referee #3:

This manuscript focuses on TCTP as an antagonist role of p53 during tumorigenesis,

We would like to add some more precision to this statement, the focus of the present study was not “on TCTP as an antagonist role of p53 during tumorigenesis”, we published that before Amson et al 2012. Actually 3 out of the 4 cell lines in Fig 2 have a deficient p53 pathway and the mice in Fig 3 as well as the cells in the rest of the paper are p53 KO. The main focus of the present paper is to examine the role of TCTP as a regulator of sEVs’ content in protein and RNA, and most importantly on the function of sEVs in the absence of TCTP in signaling of genotoxic stress and in cancer.

focusing on its effects on small EV biogenesis and intercellular communication *in vivo*.

As outlined for referee 2, nowadays’ technology cannot answer the role played by sEVs *in vivo* in mice; the field of research still requires the use of an additional *ex-vivo* step, whether for the thymocytes or the tumors originating from the double mutants. This use of a proxy in *ex-vivo* system has now been addressed in the manuscript in order to avoid any confusion or overstatement.

The authors generate a compound knockout mouse model lacking p53 combined with the inducible deletion of TCTP. The concomitant loss of TCTP and p53 results in improved overall survival and a change in the tumor spectrum such that an increased number of sarcomas are observed in this animal. The authors demonstrate that both cells derived from TCTP deficient animals have reduced numbers of sEVs, implicating this protein in EV biogenesis. In addition, TCTP does not directly bind RNA; rather, it was found to bind to DDX3, which serves as an RNA-binding protein for incorporation of certain small RNAs, such as let-7c-5p into sEVs. Chemicals that disrupt the interaction of TCTP to DDX3 are able to phenocopy the effects on mouse survival in p53 deficient animals that were observed upon combined p53 and TCTP genetic loss *in vivo*; hence, the authors propose that TCTP is critical for producing sEVs containing small RNAs that mediate the pro-tumorigenic effects of p53 loss. Although the results support a role for TCTP in modulating p53 function *in vivo*, the precise effects of TCTP-induced sEVs on these tumor-promoting phenotypes remain murky.

We don’t agree that they are “murky”.

Overall, the results lack mechanistic depth and do not fully support the conclusions that have been drawn.

We provide a genetic proof by using conditional Tctp KO mice. Fig 5, Table EV1 and Table EV4 describe the biochemical mechanism, we are not sure that this lacks mechanistic depth. In any case, we avoided any overstatement that would not fully support all the conclusions. This is also why we tried to improve our explanation that for some experiments, we had to use a proxy *ex-vivo* system since it is not feasible to provide a quantification of sEVs in mice serum.

- 1) The *in vivo* results in Fig 3 are confusing with regard to the role of sEVs. The authors demonstrate a reduction in lymphoma as well as reduced sEVs produced by thymocytes. However, a broad spectrum of sarcomas does develop in p53/TCTP double deficient mice suggesting that sarcomas can develop independently of TCTP function *in vivo*.

It is widely known that p53 KO mice develop a wide spectrum of tumors, depending on the background cross, specific mutations... There are endless discussions of lymphomas *versus* sarcomas in p53 KO, and this since the initial publications (Donehower *et al*, 1992; Harvey *et al*, 1993), there was never a satisfying explanation provided. We just noted a difference and reported it; Most probably, deletion of TCTP will induce changes drastic enough to have a change in the pathology of the tumor.

However, TCTP deletion *ex vivo* in ITR cells, a sarcoma derived from these mice, also results in reduced sEV production. In a transplant model, sEVs are able to rescue the effects of TCTP loss in ITR cells.

Are sEVs from required for the maintenance of TCTP-deficient ITR cells but not for their initiation? If so, further clarification of the role of TCTP-derived sEVs in tumor initiation vs. maintenance is needed. As it stands, the results are more confusing than illuminating, especially with regard to the role of sEVs in tumor promotion.

These ITR-1 cells (derived from a mouse sarcoma tumor) grow extremely rapidly in wild type mice. As mentioned in the Materials and Methods section, all mice were therefore sacrificed after twelve days. It is impossible in this system to discriminate between initiation and maintenance effect of sEVs. One additional point: we added sEVs from different lymphoma cell lines on ITR-1 cells deleted in TCTP. These malignant phenotypes were rescued but they remained sarcomas. This suggests that “initiation” is not influenced by sEVs, or alternatively, that a sarcoma will not become a lymphoma (data not shown).

2) The authors propose that TCTP promotes that sEV-mediated transfer of oncogenic signals, namely small RNAs, from one tumor cell to another. However, it is unclear from the results whether the sEVs rescue experiments, which employ a supraphysiological amount of sEVs (approximately 108), involve the direct transfer of specific oncogenic signals or RNA species among tumor cells. sEVs may have broader non-RNA-dependent effects on both tumor and non-tumor cells which have not been considered. Overall, the results lack mechanistic depth and do not support the conclusions that have been drawn.

We took great care to evaluate the loss of sEVs following the isolation procedure. The loss of vesicles between the EVs (100.000g pellet) and sEVs (after iodixanol gradient isolation) is +/- 28.75 fold. This is documented by Nanotrack analysis in Figure EV5 ("Evaluation of sEV yield following high-resolution iodixanol gradient"). This does not take into account the loss of vesicles during the initial sequential centrifugations to obtain the 100.000g pellet. This measurement is not feasible. We made a very conservative estimation of a 2-fold loss; it is probably more important when you search the literature. As stated in Figure EV5: "For these reasons, in the experiments using sEVs *in vitro* and *in vivo*, we used an excess of 50x of sEVs whether derived from thymocytes or cell lines" compensating hereby the loss of 28.75 fold and 2 fold mentioned above. These data indicate that we are performing our experiments in close physiological to sub-physiological conditions, not “supraphysiological”.

We report in this paper that following ablation of TCTP, we have a strong decrease of both proteins and RNA.

The referee is correct that it is probably broader, but this is outside the scope of this paper.

3) Details regarding the Cre driver are needed to interpret the in vivo model. What is the transgenic driver of Cre expression. Does the promoter effect explain why the deletion of TCTP is incomplete, not only in vivo but also when cells are treated ex vivo (e.g. Fig 4A)?

In our mice model, the Cre-Ert2 gene (Tamoxifen dependent) is inserted into the Rosa 26 locus and under the control of the Rosa 26 promoter. PCR controls are using "Rosa-Cre" primers as documented in Appendix Figure S11, but this information was originally missing from the Materials and Methods. We are sorry for that and we have corrected this lack of information.

The incomplete deletion of TCTP using an inducible system can be explained by the fact that the injected tamoxifen did not efficiently reach every tissue in the mice. This is very common in the literature.

As for ITR-1 cells, by 3 days of 4-OHT treatment (Fig 4A), TCTP is already extremely low. With further prolonged incubation, it becomes undetectable (not shown) and the cell growth is drastically decreased (Fig 4I).

4) In Fig 2, the key studies in breast cancer cell lines have been performed with a single shRNA targeting TCTP. These should be reproduced with a second shRNA to control for off-target effects.

Several shRNA were tested on human breast cancer cell lines, as documented in Fig 2A. The sh7239 was the most efficient with the different cell lines. Hence, we used only this sh7239 in Fig 2.

Moreover, the growth assays in Fig 2I should include sEV rescues from from TCTP knockdown cells to ascertain whether the effect of TCTP-deficiency is lack of sEV production or due to lack of cargo transfer (as Fig 5 later implies).

We used the ITR-1 cell system to verify the effect of sEVs derived from 4-OHT treated ITR-1 cells deleted in TCTP. This is documented in Figure EV3: Effect of sEVs from 4-OHT treated cells on 4-OHT treated cells. "The conclusion of this experiment is that even a 500x excess of sEVs derived from 4-OHT treated ITR-1 cells are unable to restore tumor cell growth."

5) The in vivo studies using sertraline in Fig 5 are low resolution in scope and do not convincingly establish a functional role for TCTP-induced sEVs in mediating the observed effects on survival.

We disagree with the referee that these are low resolution experiments. We have shown before that sertraline and thioridazine bind to TCTP, and confirmed it now using more sophisticated technology. We have published before that sertraline and thioridazine both inhibit TCTP's function. We have shown before that these drugs induce a downregulation of the expression of TCTP.

Is the tumor spectrum similar to that observed with the double knockout mice in Fig 3. What is the effect of sertraline on sEV production from p53 deficient thymocytes?

While we show in the manuscript that the treatment of the p53KO mice with sertraline has a similar effect on survival as the genetic ablation of *tctp* in the double mutants, we did not look at the tumor spectrum in this experiment or at the effect of sertraline on sEVs production as by using a pharmacological compound, we thought that it might not be as reliable.

References

- Amson R, Pece S, Lespagnol A, Vyas R, Mazzarol G, Tosoni D, Colaluca I, Viale G, Rodrigues-Ferreira S, Wynendaele J *et al* (2012) Reciprocal repression between P53 and TCTP. *Nat Med* 18: 91-99
- Amson R, Pece S, Marine JC, Di Fiore PP, Telerman A (2013) TPT1/ TCTP-regulated pathways in phenotypic reprogramming. *Trends Cell Biol* 23: 37-46
- Azzam EI, de Toledo SM, Gooding T, Little JB (1998) Intercellular communication is involved in the bystander regulation of gene expression in human cells exposed to very low fluences of alpha particles. *Radiat Res* 150: 497-504
- Azzam EI, Little JB (2004) The radiation-induced bystander effect: evidence and significance. *Hum Exp Toxicol* 23: 61-65
- Banroques J, Cordin O, Doere M, Linder P, Tanner NK (2008) A conserved phenylalanine of motif IV in superfamily 2 helicases is required for cooperative, ATP-dependent binding of RNA substrates in DEAD-box proteins. *Mol Cell Biol* 28: 3359-3371
- Banroques J, Cordin O, Doere M, Linder P, Tanner NK (2011) Analyses of the functional regions of DEAD-box RNA "helicases" with deletion and chimera constructs tested in vivo and in vitro. *J Mol Biol* 413: 451-472
- Bommer UA, Telerman A (2020) Dysregulation of TCTP in Biological Processes and Diseases. *Cells* 9
- Crescitelli R, Lässer C, Lötval J (2021) Isolation and characterization of extracellular vesicle subpopulations from tissues. *Nat Protoc* 16: 1548-1580
- Donehower LA, Harvey M, Slagle BL, McArthur MJ, Montgomery CA, Jr., Butel JS, Bradley A (1992) Mice deficient for p53 are developmentally normal but susceptible to spontaneous tumours. *Nature* 356: 215-221
- Gyuris A, Navarrete-Perea J, Jo A, Cristea S, Zhou S, Fraser K, Wei Z, Krichevsky AM, Weissleder R, Lee H *et al* (2019) Physical and Molecular Landscapes of Mouse Glioma Extracellular Vesicles Define Heterogeneity. *Cell Rep* 27: 3972-3987 e3976
- Hagiwara K, Katsuda T, Gailhouste L, Kosaka N, Ochiya T (2015) Commitment of Annexin A2 in recruitment of microRNAs into extracellular vesicles. *FEBS Lett* 589: 4071-4078
- Harvey M, McArthur MJ, Montgomery CA, Jr., Bradley A, Donehower LA (1993) Genetic background alters the spectrum of tumors that develop in p53-deficient mice. *FASEB J* 7: 938-943
- Jeppesen DK, Fenix AM, Franklin JL, Higginbotham JN, Zhang Q, Zimmerman LJ, Liebler DC, Ping J, Liu Q, Evans R *et al* (2019) Reassessment of Exosome Composition. *Cell* 177: 428-445 e418

- Li S, Chen M, Xiong Q, Zhang J, Cui Z, Ge F (2016) Characterization of the Translationally Controlled Tumor Protein (TCTP) Interactome Reveals Novel Binding Partners in Human Cancer Cells. *J Proteome Res* 15: 3741-3751
- Mateescu B, Kowal EJ, van Balkom BW, Bartel S, Bhattacharyya SN, Buzás EI, Buck AH, de Candia P, Chow FW, Das S *et al* (2017) Obstacles and opportunities in the functional analysis of extracellular vesicle RNA - an ISEV position paper. *J Extracell Vesicles* 6: 1286095
- Senissar M, Le Saux A, Belgareh-Touzé N, Adam C, Banroques J, Tanner NK (2014) The DEAD-box helicase Ded1 from yeast is an mRNP cap-associated protein that shuttles between the cytoplasm and nucleus. *Nucleic Acids Res* 42: 10005-10022
- Sirois I, Raymond MA, Brassard N, Cailhier JF, Fedjaev M, Hamelin K, Londono I, Bendayan M, Pshezhetsky AV, Hebert MJ (2011) Caspase-3-dependent export of TCTP: a novel pathway for antiapoptotic intercellular communication. *Cell Death Differ* 18: 549-562
- Takahasi K, Yoneyama M, Nishihori T, Hirai R, Kumeta H, Narita R, Gale M, Jr., Inagaki F, Fujita T (2008) Nonsel self RNA-sensing mechanism of RIG-I helicase and activation of antiviral immune responses. *Mol Cell* 29: 428-440
- Villarroya-Beltri C, Gutierrez-Vazquez C, Sanchez-Cabo F, Perez-Hernandez D, Vazquez J, Martin-Cofreces N, Martinez-Herrera DJ, Pascual-Montano A, Mittelbrunn M, Sanchez-Madrid F (2013) Sumoylated hnRNP A2B1 controls the sorting of miRNAs into exosomes through binding to specific motifs. *Nat Commun* 4: 2980
- Zhang H, Freitas D, Kim HS, Fabijanic K, Li Z, Chen H, Mark MT, Molina H, Martin AB, Bojmar L *et al* (2018) Identification of distinct nanoparticles and subsets of extracellular vesicles by asymmetric flow field-flow fractionation. *Nat Cell Biol* 20: 332-343

Dear Prof. Telerman,

Thank you for the submission of your revised manuscript to our editorial offices. I have already forwarded the report from the referee that I asked to re-evaluate your study, you will find again below. I also have received your point-by-point-response (further revision plan). The referee has remaining concerns, and also feels that some of his/her points have not been adequately addressed, indicating that the paper needs further revision. After looking through your revision plan and further cross-commenting with the referee, and also considering that original referee #1 fully supported publication of the study, I decided to invite a final revised manuscript that addresses the remaining referee points as indicated in your revision plan. Please also provide a detailed final point-by-point-response to these.

- We now use CRediT to specify the contributions of each author in the journal submission system. CRediT replaces the author contribution section. Please use the free text box to provide more detailed descriptions and do NOT provide your final manuscript text file with an author contributions section. See also our guide to authors: <https://www.embopress.org/page/journal/14693178/authorguide#authorshipguidelines>
- The Data Availability section should only contain information on large datasets that have been deposited to external repositories and all access information. Please remove the statement: 'All data needed to evaluate the paper's conclusions are present.' If no datasets have been deposited for this study, please state here: 'No primary datasets have been generated and deposited'. Finally, please name this section 'Data availability section'.
- Please remove the section "Permits" from the manuscript text file and also remove information on guidelines and regulations from the Disclosure and Competing Interests Statement. Please move these information to an ethics statement (or a statement named 'animal experiments') in the methods section.
- Please order the manuscript sections like this, using these names:
Title page - Abstract - Keywords - Introduction - Results - Discussion - Materials and Methods - Data availability section - Acknowledgements - Disclosure and Competing Interests Statement - References - Figure legends - Expanded View Figure legends
- Could the last three EV figures be combined to have one Fig. EV2? I think this would render the manuscript more comprehensive and easier to typeset. In case you decide to combine the panels, please carefully update all the callouts.
- Please add scale bars of similar style and thickness to all the microscopic images, using clearly visible black or white bars (depending on the background). Please place these in the lower right corner of the images themselves. Please do not write on or near the bars in the image but define the size in the respective figure legend. Presently, there are some images with scale bars with (hardly readable) text nearby.
- Please make sure that the number "n" for how many independent experiments were performed, their nature (biological versus technical replicates), the bars and error bars (e.g. SEM, SD) and the test used to calculate p-values is indicated in the respective figure legends (for main, EV and Appendix figures) of the final revised manuscript. Please also check that all the p-values are explained in the legend, and that these fit to those shown in the figure. Please provide statistical testing where applicable. Please avoid the phrase 'independent experiment', but clearly state if these were biological or technical replicates. Please also indicate (e.g. with n.s.) if testing was performed, but the differences are not significant. In case n=2, please show the data as separate datapoints without error bars and statistics. See also:
<http://www.embopress.org/page/journal/14693178/authorguide#statisticalanalysis>
- If n<5, please show single datapoints for diagrams. Could statistics also be added to the diagram in Fig. 3F and in the Appendix? Moreover, although 'n' is provided, please describe the nature of entity for 'n' (technical or biological) in the legends of figures 2d-f; 2h-i; 4d-f; 4h-i; 5c; 5h; 5m; EV 3a.
- Please make sure that all figure panels are called out separately and sequentially. Presently, there seems to be no callout for Fig 3G. Moreover, a 'Supplementary Table 2; is mentioned, but doesn't exist. Please check.
- Tables EV1-EV4 are datasets. Please name these Datasets EV1-EV4 and upload the original excel files as datasets, with a legend and a title on the first TAB. Please remove their legends from the manuscript text file and update their callouts (Dataset EVx).
- Please make sure that all the funding information is also entered into the online submission system and that it is complete and similar to the one in the acknowledgement section of the manuscript text file. Presently, grants Science without Borders (CNPq),

Federal University of Parana (UFPR, Brazil), MESR, and Ecole Doctorale de Cancérologie Paris XI are missing in the submission system.

- During our standard image analysis, we detected potential aberrations in the Appendix figure set, and we would like to clarify these issues: Please provide source data (uncropped images) for the Western blots and gels shown in the Appendix, in particular for Fig. S1E and S1I.

In addition, I would need from you:

Yours sincerely,

Referee #2:

Some parts of the manuscript have improved, but there are still serious flaws that have not been sufficiently addressed:

- The explanation of why the authors can't quantify sEVs in the sera of these mice is robust and now suitably described in the transcript. They've also shown that sEVs from 4-OHT treated mice could not restore the growth rate (ATP production) of TCTP-deficient mice in the same way that sEVs from TCTP-competent cells can, which they lacked previously.
- However, as they looked at the pro-malignant function of TCTP in small extracellular vesicles, specifically, is it not possible that TCTP loss could promote the secretion of smaller/larger/other specific vesicle populations and that these could be equally pro-malignant? If they showed that EVs from TCTP-competent mice only enhanced the proliferation of TCTP-deficient mice and that a specific population of sEVs were the vesicles underlying this behaviour this manuscript would be significantly more impactful, even if figure 5 was completely omitted.
- If the purpose of the paper isn't to identify the functionally responsible RNAs it is unclear why the authors want to include Figure 5, particularly as the data is really weak and a bit contradictory (e.g. 5J 0 sertraline can revert some of the survival of mice, but it is not impacting on binding of let7c-5p to DDX3/TCTP in figure 5L suggesting the rescue by sertraline in 5J is TCTP independent)
- The explanation of why figure 1 was included is not particularly strong. I suppose for the broader community it's good to have more data out there than less but demonstrating the pro-apoptotic role of TCTP in normal cells seems unnecessary for a paper investigating the pro-malignant role of TCTP in tumour cells
- The authors suggest "it is technically impossible to discriminate cellular TCTP from sEV's" and that they can't therefore see what happens with TCTP in sEVs once it reaches the cell that ultimately internalises it. A simple experiment to prove this would be to generate sEVs from a cell line with tagged TCTP and applying them to cells without labelled TCTP...
- The authors state that "inhibiting TCTP by sertraline or thioridazine has, as a consequence, a decreased binding of DDX3 to the RNA (please see Figure 5)" despite the fact that figure 5 outright refutes the capacity of sertraline to reduce the binding of DDX3 to miR-let7, and may not convincingly show that thioridazine does either (no loading control in lane so can't see if the lane is loaded appropriately).
- The fact that the data in Fig5 is done with purified proteins and doesn't test whether the drugs affect sEV production isn't a justification, rather the authors just acknowledge a major flaw in their paper. Further, the only functional data they show for the drugs is that sertraline improved the survival of tp53^{-/-} mice but the figure then goes on to refute the idea that sertraline can in fact reduce DDX3-let7 binding suggesting this functions by an entirely different mechanism, and as they didn't test the impact of sertraline to enhance survival of TCTP knockout mice (which they had access to) it's impossible to say that it is in any way TCTP-dependent.
- It is a fair conclusion for the authors to suggest that investigation into the RNAs would warrant a future project but is beyond the scope of this work. However, by touching on RNAs at all it needs to be far more convincing that something is actually going on here or that RNAs in some way contribute to any of the previous findings.

We thank referee #2 for her/his time and comments. Please see below our answer in blue.

Referee #2

Some parts of the manuscript have improved, but there are still serious flaws that have not been sufficiently addressed:

- The explanation of why the authors can't quantify sEVs in the sera of these mice is robust and now suitably described in the transcript. They've also shown that sEVs from 4-OHT treated mice could not restore the growth rate (ATP production) of TCTP-deficient mice in the same way that sEVs from TCTP-competent cells can, which they lacked previously.

We are pleased to have satisfactorily addressed the questions raised by the reviewer. This was a primary concern and we have taken great care in addressing it. All the major points (1 to 9), which were previously identified as the “weakest point of this manuscript”, have been explained and corrected in the paper.

There is one additional important piece of information that will hopefully please referee 2. She/he was so suspicious of the results presented in Figure 5.

As shown in Figure 5A-C and expanded in EV Table 1, a series of microRNAs were pulled down by a CLIP with anti-TCTP. We analyzed a public database of the microRNAs binding DDX3 (Huang et al, 2022). Out of the 128 microRNAs binding to DDX3 (Huang et al., 2022), 43 of them were found in our anti-TCTP CLIP (Dataset EV5). This is very informative and suggests that TCTP is recruiting these microRNAs via DDX3. We included these data at the end of the rebuttal letter. These 43 microRNAs are highlighted in yellow in the attached new Excel file containing the 128 microRNAs that bind DDX3X.

This was added in the main text (results section: The mechanism by which TCTP regulates the content of miRNAs in sEVs is DDX3-dependent):

To investigate whether at least some of the microRNAs pulled down by the anti-TCTP CLIP are also binding DDX3, we analyzed a public database of the microRNAs binding DDX3 (Huang et al., 2022). Out of the 128 microRNAs binding to DDX3 (Huang et al., 2022), 43 of them were found in our anti-TCTP CLIP (Dataset EV1). This suggests that TCTP is recruiting these microRNAs via DDX3. These 43 microRNAs are highlighted in yellow (Dataset EV5).

This was added to the discussion:

Of note, out of the 128 microRNAs binding to DDX3 (Huang et al., 2022), (Dataset EV5), 43 of them were found in our anti-TCTP CLIP.

- However, as they looked at the pro-malignant function of TCTP in small extracellular vesicles, specifically, is it not possible that TCTP loss could promote the secretion of smaller/larger/other specific vesicle populations and that these could be equally pro-malignant?

This is a new question: the reviewer is asking whether the deletion of TCTP could translocate oncogenic/transforming vesicular signaling to other vesicles that would not be the sEVs. It is very hard to imagine how and why such an event would occur. The sorting of proteins and RNAs into the sEVs is a tightly controlled process, and the content of sEVs, with thousands of proteins and RNAs, cannot “jump” or be rerouted to other vesicles. This is, at least, with today’s knowledge of this kind of trafficking.

The data below were added in Fig EV 3C.

Complementation experiments (in vitro): Effect of EVs (sEVs + NVs) from 4-OHT treated cells on the growth of 4-OHT treated cells. Growth of ITR-1 cells before (red), after treatment with 4-OHT (grey) and the latter following addition of EVs from 4-OHT treated ITR-1 cells (orange). For 20,000 cells cultured, 17×10^7 EVs were added (independent biological replicates $n=4$). Data information: Statistical analysis was performed using Mann-Whitney test Mean \pm SEM. ns (not significant) *** $P < 0.001$.

Despite the complexity of the question, we have a clear answer for that. Before working on sEVs we conducted some preliminary experiments with EVs. As explained in the previous rebuttal letter, these EVs contain the vesicular and non-vesicular material secreted by the cells. Our experiments clearly show that EVs from TCTP $-/-$ tumor cells cannot restore the malignant growth.

This was added in the main text (results section: sEV-regulated malignant transformation signaling is Tctp-dependent):

Neither EVs nor sEVs derived from 4-OHT treated ITR-1 cells, thus depleted of TCTP, could restore the malignant growth (Fig EV 3A-C).

If they showed that EVs from TCTP-competent mice only enhanced the proliferation of TCTP-deficient mice and that a specific population of sEVs were the vesicles underlying this behaviour this manuscript would be significantly more impactful, even if figure 5 was completely omitted.

As shown in the data provided above, when TCTP is deleted, there is no vesicular or non-vesicular material that could compensate for the malignant growth. There is a logic behind this: TCTP is present in the sEVs, so its deletion would not induce oncogenic signaling in other vesicles.

It is not currently possible, with the knowledge and technology available nowadays, to isolate within the sEVs the population that would be carrying this oncogenic signaling. As explained already, it is quite tedious to obtain a pure population of sEVs using high-resolution iodixanol gradients fractionation. To start identifying within this population a subpopulation responsible for malignant growth and using such vesicles for biological experiments is not feasible with today's status of the methodologies. Such subpopulations of sEVs have not been clearly characterized besides for the expression of a series of proteins by Western Blot analysis. We believe that we have pushed this part of the study as far as it is feasible nowadays.

This was further added in the discussion: Since the question was raised that following deletion of Tctp there could be a potential pro-oncogenic signaling that would shift from the initially observed sEVs to other EVs, we investigated whether EVs (containing both the NVs and sEVs) derived from 4-OHT treated ITR-1 cells (depleted in Tctp) could rescue the malignant growth; we did not observe any rescue.

- If the purpose of the paper isn't to identify the functionally responsible RNAs it is unclear why the authors want to include Figure 5, particularly as the data is really weak and a bit contradictory (e.g. 5J 0 sertraline can revert some of the survival of mice, but it is not impacting on binding of let7c-5p to DDX3/TCTP in figure 5L suggesting the rescue by sertraline in 5J is TCTP independent)

The only purpose of this paper is, as stated in the title, to show that TCTP regulates intercellular vesicular signaling in genotoxic stress and tumorigenicity. However, in each system that we used, we show that conditional deletion of TCTP, or knocking down TCTP has, as consequence, a decrease in protein and RNA in the sEVs. Thus, we tried to understand the potential link between TCTP and RNAs in more detail (Figure 5). For that purpose, we started with a CLIP using anti-TCTP antibodies (5A). These experiments show that, indeed, TCTP forms complexes with RNAs, and we focused more specifically on miRNAs. Fig 5B provides a shortlist of some of the miRNAs and shows validation by qRT-PCR in sEVs (Fig 5 B).

These results are in line with the hypothesis that TCTP forms complexes with RNAs and provide an explanation as to why a decrease in TCTP has as consequence a decrease in RNA. In Figures 5D and E, we show that this binding between TCTP and RNA is most probably indirect. This is completely novel and contradicts the finding that TCTP was probably a direct RNA binding protein.

So, we went for an alternative hypothesis, namely that TCTP is binding an RNA binding protein. We explained in the previous rebuttal letter and in the paper why we investigated the binding of the helicase DDX3 to TCTP. DDX3, since it is a DEAD box helicase, is binding an exhaustive list of RNAs including miRNAs, as reported in the miRTarBase data base.

We show that TCTP binds DDX3 with an affinity (K_D) of 68.9 nM

We show that TCTP-DDX3 and Let-7c-5p form complexes

We show that TCTP potentiates the recruitment of Let-7c-5p by DDX3

Hence, we conclude that TCTP binds specifically to DDX3 and helps recruiting Let-7c-5p (Let-7c-5p was used as an example).

It is very likely that TCTP potentially forms complexes with other RNA binding proteins (Li et al, 2016), but it was not the purpose of this paper to investigate all of them.

To answer specifically your statement: “If the purpose of the paper isn't to identify the functionally responsible RNAs”. As you know in the shortlist provided in Figure 5B and certainly in the EV Table 1, there are a handful of miRNAs able to act as oncogenes or tumor suppressors. This is a coordinated biological response and most probably not the work of a single miRNA. While specific papers single out miRNAs in their oncogenic function, it is widely known that as little as one miRNA is potentially able to regulate the expression of hundreds of genes, and one transcript can be affected by multiple miRNAs (Lewis et al, 2005; O'Day & Lal, 2010). Some of them are directly related to DDX3 (Ni et al, 2020; Zhao et al, 2016).

Although the topic may be of interest, it is important to acknowledge that it is not the intended focus of this paper. Our focus is to assess the regulation of intercellular vesicular signaling in genotoxic stress and tumorigenicity by TCTP. There is nothing “really weak and a bit contradictory” in these figures. Fig 5G is highly significant for the survival of p53KO mice treated with sertraline, which inhibits, as shown before, the function of TCTP. Both sertraline and thioridazine disrupt the binding of DDX3 to TCTP, but not entirely...This is biology, and hopefully, the next generation of anti-TCTP compounds will be more efficient. As for 5L and M, these are the most representative and reproducible experiments. Clearly sertraline is less efficient than thioridazine in disrupting TCTP-DDX3-Let-7c-5p. These are in vitro experiments, which are indicative of a trend. We do not wish to hide or remove them. Otherwise, the reader would be left with the impression that both drugs disrupt TCTP-DDX3 as in Figure 5K. The matter is considerably intricate and would require a comprehensive approach. It is essential to acknowledge that not everything can be resolved with a single paper.

We added the following sentences to the discussion:

Altogether, the data presented indicate that deletion of Tctp has as a consequence a decreased number of sEVs secreted per cell, concomitant with a decrease in protein and RNA content. At least part of the transport of miRNAs by TCTP is mediated as a complex with DDX3. Many of these miRNAs could affect cell growth. However, it is important to take into consideration the fact that in the sEVs, many of these miRNAs are present together with other RNA species and proteins, some of which are RNA binding proteins, suggesting that the outcome of the phenotype could be a coordinated response of these different RNAs and proteins (Lewis et al., 2005; O'Day & Lal, 2010). Some of them are directly related to DDX3 (Ni et al., 2020; Zhao et al., 2016).

- The explanation of why figure 1 was included is not particularly strong. I suppose for the broader community it's good to have more data out there than less but demonstrating the pro-apoptotic role of TCTP in normal cells seems unnecessary for a paper investigating the pro-malignant role of TCTP in tumour cells

We consider it appropriate to start a research paper with a new genetic model by describing the effect of this conditional ablation of TCTP in normal cells rather than to start directly with cancer. By outlining the effects of this model, we gain a deeper understanding of its implications. Moreover, the “Bystander effect” has been investigated for many years without

providing a molecular or genetic answer. Our intent was not to be reaching a “broader community” nor to “have more data out there than less”.

- The authors suggest "it is technically impossible to discriminate cellular TCTP from sEV's" and that they can't therefore see what happens with TCTP in sEVs once it reaches the cell that ultimately internalises it. A simple experiment to prove this would be to generate sEVs from a cell line with tagged TCTP and applying them to cells without labelled TCTP...

This was addressed in the initial rebuttal letter in answer to “major comments # 6”:

“Major comments # 6”: “The vesicle isolation suggest TCTP is expressed in the vesicles. What happens when the vesicles are imported, do the cells that takes these up use this TCTP? Can an increased TCTP be seen in Figure 4M, similar to 4K. Would be good to show TCTP levels here”.

The following answer to “Major comments # 6” was provided below:

Unfortunately, it is technically impossible to discriminate the cellular TCTP from sEVs’ TCTP in full-grown tumors. In addition, the discrimination between the initial amount of TCTP present in the supplemented sEVs and the amount of TCTP in the tumors would be elusive, as tumor growth occurred in a WT microenvironment (WT mice injected with ITR-1 cells supplemented with sEVs) (Fig 4 M).

The figure below was added now in the manuscript as (Appendix Fig S6):

TCTP expression in ITR-1 cells upon supplemented with sEVs.

Western blot analysis of Tctp expression in untreated ITR-1 cells (1) and in 4-OHT treated ITR-1 cells (2) supplemented with sEVs from ITR-1 cells (3). Gapdh was used as loading control. Quantification of the bands (Image Lab software from Bio-Rad).

The ITR-1 cells have been engineered as conditional TCTP knockdown cells. Before treatment with 4-OH Tamoxifen (4-OHT), these cells are full of TCTP. In order to “generate sEVs from a cell line with tagged TCTP”, this would be the equivalent of over-expressing TCTP in ITR-1 cells that already express it very highly. We do not favor overexpression experiments; this could change the behavior of these cells that are already growing very fast, and overexpression could lead to a misinterpretation of the data. This is why we checked the uptake of FITC labeled sEVs. We can see an increase in TCTP expression in ITR-1 cells deleted in TCTP and supplemented with sEVs from parental not treated sEVs (see above). We never see a drastic increase in TCTP but clearly sufficient to promote malignant growth. These vesicles contain significant information imported, due to the function of TCTP.

We added in the results section “sEV-regulated malignant transformation signaling is Tctp-dependent”: We can observe an increase in the expression of TCTP in ITR-1 cells that were treated with 4-OHT and supplemented with sEVs from untreated parental ITR-1 cells (Appendix Fig S6). These findings suggest that the sEVs do uptake TCTP, which is

consistent with the results obtained from the experiments involving FITC-labeled sEVs (Fig 4G and H).

This was already commented in the Discussion:

Hence, the complementation experiments using the inducible Tctp-knockout cell line ITR-1, showed that these Trp53^{-/-} cells that lost their malignancy as a result of deleting both Tctp alleles, regained it upon supplementation with sEVs from the malignant parental ITR-1 cell line (Fig 4).

- The authors state that "inhibiting TCTP by sertraline or thioridazine has, as a consequence, a decreased binding of DDX3 to the RNA (please see Figure 5)" despite the fact that figure 5 outright refutes the capacity of sertraline to reduce the binding of DDX3 to miR-let7, and may not convincingly show that thioridazine does either (no loading control in lane so can't see if the lane is loaded appropriately).

We have already addressed these issues: "Clearly sertraline is less efficient than thioridazine in disrupting TCTP-DDX3-Let-7c-5p". We take full responsibility for the confusion caused and apologize for that. The problem is not in our paper, but in the interpretation of these experiments by the referee, probably because we did not explain this adequately in the first rebuttal. In the result section of the manuscript, it is written: "Given the above results, we investigated the effects of sertraline and thioridazine on the TCTP-DDX3 complexes. Thioridazine, and to a lesser degree sertraline, disrupted TCTP-DDX3 complex formation (Fig 5K). We then tested the complex formation between TCTP-DDX and let-7c-5p (Fig 5L) in the presence or absence of each of the pharmacological agents. Sertraline did not display any significant effect. On the other hand, thioridazine disrupted almost entirely the complex formation (Fig 5L and M)". Figure 5 does not "refute the capacity of sertraline to reduce the binding", but shows a decreased binding of DDX3 to the RNA, but not significant.

Figure 5L is not a Western blot but testing the formation of protein-RNA complexes. This is a non-denaturing gel transferred on Amersham HybondTM-N+ membranes in order to detect labeled RNA (here let7c-5p) using the LightShiftTM RNA EMSA Optimization (Thermo Scientific 20158X). The only labeled molecule in this experiment (Figure 5L) is Let-7c-5p. As shown in Figure 5L, free Let-7c-5p is present in all lanes at the bottom of the gel. We are not dealing with cell extracts, just with 2 recombinant proteins DDX3 and TCTP adding a labeled microRNA: Let-7c-5p. If there is enough of the labeled RNA so that we could see a shift (representative of a complex formation) appearing or disappearing on a gel, we are all set. One would need a loading control if the microRNA (here Let-7c-5p) is not visible and then use ethidium bromide, here there is plenty of Let-7c-5p and the free/ unbound microRNA is seen as the fat overexposed band at the bottom of the gel.

For Western Blots throughout this paper, we have always used equal loading.

The only conclusions are that TCTP-DDX3-Let-7c-5p form complexes. TCTP-DDX3 are disrupted by sertraline and thioridazine. TCTP-DDX3-Let-7c-5p is completely disrupted by thioridazine. We used the same incubation condition for both sertraline and thioridazine. In preliminary pilot experiments, we tested whether a prolonged incubation time with sertraline would result in further disruption of the complex, which was indeed observed. However, while this was an interesting point, we did not think that such an experimental setup would lead to the production of reliable results, as these would not be comparable conditions between drug treatments and would not be truthful with the fact that thioridazine is more disruptive to the complex.

- The fact that the data in Fig5 is done with purified proteins and doesn't test whether the drugs affect sEV production isn't a justification, rather the authors just acknowledge a major flaw in their paper.

Again, we apologize for the confusion caused by insufficient clarity regarding our data. We initially found that thioridazine and sertraline kill tumor cells and significantly decrease the level of TCTP (Tuynder et al, 2004). We later found that both drugs are binding TCTP and inhibit its function (Amson et al, 2012). . These results were reproduced by other groups (Baldissera et al, 2023; Boia-Ferreira et al, 2017). We also found that sertraline kills ex-vivo AML patient cells (Amson et al, 2013). Both drugs induce, after 24 to 48 hours, a drastic decrease in TCTP, and they kill the tumor cells.

Indeed, both drugs affect the function of sEVs, and there are many explanations for that. They disrupt the binding of TSAP6 to TCTP, and they inhibit the function of TCTP itself, with all the consequences on the biology of sEVs as addressed in this paper. This is true in vitro using these drugs. In vivo we would not be able to distinguish whether the decrease in sEVs would be due only to a direct effect or to cells that are simply perturbed and dying with many biological processes being disrupted. Any experiment where sEVs alone would be incubated with both drugs could be completely artefactual, and we rather avoid this. Such drugs would be toxic to the sEVs and we would not be able to conclude anything. Hence there is no "major flaw in the paper", those are complicated issues, and we therefore generated the genetic models.

Further, the only functional data they show for the drugs is that sertraline improved the survival of tp53^{-/-} mice but the figure then goes on to refute the idea that sertraline can in fact reduce DDX3-let7 binding suggesting this functions by an entirely different mechanism, and as they didn't test the impact of sertraline to enhance survival of TCTP knockout mice (which they had access to) it's impossible to say that it is in any way TCTP-dependent.

The survival curve of the conditional TCTP knockout mice (Figure 3 in green) is similar to the survival of normal wild type mice. These conditional TCTP knockout mice do not form tumors and are not dying within this period. This is the same as for wild type mice, as widely reported in the literature. Indeed, we "didn't test the impact of sertraline to enhance survival of TCTP knockout mice" because they have a survival equivalent to normal wild type mice. Of note, we do not see the scientific justification for treating tumor-free TCTP-deficient mice with a TCTP inhibitor.

- It is a fair conclusion for the authors to suggest that investigation into the RNAs would warrant a future project but is beyond the scope of this work. However, by touching on RNAs at all it needs to be far more convincing that something is actually going on here or that RNAs in some way contribute to any of the previous findings.

We have already addressed this issue at the beginning of this rebuttal letter: in each system that we used, we show that conditional deletion of TCTP, or knocking down TCTP, has, as a consequence, a decrease in protein and RNA in the sEVs. The data in Figure 5, show that anti-TCTP CLIP pulls down a large series of microRNAs. Along this, the qRT-PCR analysis confirms that at least part of these microRNAs are found in sEVs in a TCTP dependent manner, which is completely novel and informative. The fact that TCTP does not bind directly

RNAs but in complex with other carriers as shown in Figure 5 for DDX3, and that ultimately out of the 128 microRNAs binding to DDX3, 43 of them were found in our anti-TCTP CLIP seems important enough to be included in this work.

References

- Amson R, Karp JE, Telerman A (2013) Lessons from tumor reversion for cancer treatment. *Curr Opin Oncol* 25: 59-65
- Amson R, Pece S, Lespagnol A, Vyas R, Mazzarol G, Tosoni D, Colaluca I, Viale G, Rodrigues-Ferreira S, Wynendaele J *et al* (2012) Reciprocal repression between P53 and TCTP. *Nat Med* 18: 91-99
- Baldissera AB, Boia-Ferreira M, Basílio ABC, Resende JSS, Castro MAA, Chaim OM, Gremski LH, Veiga SS, Senff-Ribeiro A (2023) Sertraline as a potential cancer therapeutic approach: Biological relevance of TCTP in breast cancer cell lines and tumors. *Adv Med Sci* 68: 227-237
- Boia-Ferreira M, Basílio AB, Hamasaki AE, Matsubara FH, Appel MH, Da Costa CRV, Amson R, Telerman A, Chaim OM, Veiga SS *et al* (2017) TCTP as a therapeutic target in melanoma treatment. *Br J Cancer* 117: 656-665
- Huang HY, Lin YC, Cui S, Huang Y, Tang Y, Xu J, Bao J, Li Y, Wen J, Zuo H *et al* (2022) miRTarBase update 2022: an informative resource for experimentally validated miRNA-target interactions. *Nucleic Acids Res* 50: D222-D230
- Lewis BP, Burge CB, Bartel DP (2005) Conserved seed pairing, often flanked by adenosines, indicates that thousands of human genes are microRNA targets. *Cell* 120: 15-20
- Li S, Chen M, Xiong Q, Zhang J, Cui Z, Ge F (2016) Characterization of the Translationally Controlled Tumor Protein (TCTP) Interactome Reveals Novel Binding Partners in Human Cancer Cells. *J Proteome Res* 15: 3741-3751
- Ni J, Chen L, Ling L, Wu M, Ren Q, Zhu W (2020) MicroRNA-196a promotes cell proliferation and inhibits apoptosis in human ovarian cancer by directly targeting DDX3 and regulating the PTEN/PI3K/AKT signaling pathway. *Mol Med Rep* 22: 1277-1284
- O'Day E, Lal A (2010) MicroRNAs and their target gene networks in breast cancer. *Breast Cancer Res* 12: 201
- Tuynder M, Fiucci G, Prieur S, Lespagnol A, Geant A, Beaucourt S, Duflaut D, Besse S, Susini L, Cavarelli J *et al* (2004) Translationally controlled tumor protein is a target of tumor reversion. *Proc Natl Acad Sci U S A* 101: 15364-15369
- Zhao L, Mao Y, Zhao Y, He Y (2016) DDX3X promotes the biogenesis of a subset of miRNAs and the potential roles they played in cancer development. *Sci Rep* 6: 32739

Prof. Adam Telerman
Institut Gustave Roussy
U981
114, rue Édouard-Vaillant
Villejuif, Ile de France 94805
France

Dear Prof. Telerman,

Thank you for the submission of your final revised manuscript to our editorial offices. I now went through your final p-b-p-response and the manuscript and, also as I have discussed the revision plan already with the referee, consider the remaining concerns sufficiently addressed.

I am thus pleased to accept your manuscript for publication in the next available issue of EMBO reports. Thank you for your contribution to our journal.

Yours sincerely,
